computational mechanics/structural engineering/ fluid mechanics

fluid–structure interaction, finite-element method, dissipation, symbolic regression, functional relation

**Authors for correspondence:**
Rajdeep Dutta
e-mail: rajdeep_dutta@i2r.a-star.edu.sg
Saikat Sarkar
e-mail: saikat@iiti.ac.in

# Capturing functional relations in fluid–structure interaction via machine learning

Tejas Soni[1,†], Ashwani Sharma[1,†], Rajdeep Dutta[2], Annwesha Dutta[3,4], Senthilnath Jayavelu[2] and Saikat Sarkar[1]

[1]Department of Civil Engineering, Indian Institute of Technology Indore, Indore, Madhya Pradesh, India
[2]Department of Machine Intellection, Institute for Infocomm Research Technology and Research Agency for Science, Singapore, Singapore
[3]ICTP - The Abdus Salam International Centre for Theoretical Physics, Strada Costiera 11, Trieste 34151, Italy
[4]Department of Physics, Indian Institute of Science Education and Research, Tirupati 517507, India

  RD, 0000-0003-0195-2153; SS, 0000-0002-5858-3179

While fluid–structure interaction (FSI) problems are ubiquitous in various applications from cell biology to aerodynamics, they involve huge computational overhead. In this paper, we adopt a machine learning (ML)-based strategy to bypass the detailed FSI analysis that requires cumbersome simulations in solving the Navier–Stokes equations. To mimic the effect of fluid on an immersed beam, we have introduced dissipation into the beam model with time-varying forces acting on it. The forces in a discretized set-up have been decoupled via an appropriate linear algebraic operation, which generates the ground truth force/moment data for the ML analysis. The adopted ML technique, symbolic regression, generates computationally tractable functional forms to represent the force/moment with respect to space and time. These estimates are fed into the dissipative beam model to generate the immersed beam's deflections over time, which are in conformity with the detailed FSI solutions. Numerical results demonstrate that the ML-estimated continuous force and moment functions are able to accurately predict the beam deflections under different discretizations.

# 1. Introduction

With the increase in computational power, fluid–structure interaction (FSI) has gained significant interest in a broad range of applications [1]. FSI phenomena is omnipresent in biomechanics

†These authors have contributed equally to this study.

to engineering problems, such as blood flowing through arteries, aeroplanes flying in the air and submarines sailing through the sea, where fluid motions anchor the associated structural dynamics [2–8]. In marine and aviation engineering, there exist multi-billion dollar industries that solely depend on day-to-day advancements in FSI technologies and efficient methods to maintain their economies. FSI technologies have tremendous potential in underwater construction and civil engineering projects as well [9], and a slight improvement in FSI models and their computational aspects can be highly economic. Modern cities demand keeping power transmission cables out of sight by installing them underground and in order to do so, these cables often pass through water. In these structural designs, one needs to keep in mind the indispensability factor and prevent superfluous use of materials for cost-effectiveness. Hence, to achieve a robust yet economic solution to FSI problems, it demands a precise understanding of, (i) how fluid dynamics and structural dynamics affect each other? (ii) which one plays a dominant role depending on a use case? and (iii) what are the available inputs that can be tuned to drive the interaction dynamics in a guided manner?

To study the behaviour of a structure while interacting with a fluid, many theories have been proposed since the term FSI was coined. The immersed boundary (IB) method was developed by Charles Peskin [10] in 1972 in the field of biomechanics to study the nature of blood flow through the heart. This study realized the interaction between a viscous, incompressible fluid and a flexible structure submerged inside it, and performed a complete computational analysis of the underlying interaction. The novelty lay in the modelling of a fully coupled FSI problem, which involves complex time-dependent geometries by discretizing the fluid domain on an Eulerian mesh and the immersed structure on a Lagrangian grid. Peskin's method [10,11] exploits the Dirac-delta function to transfer force from IB to fluid, and velocity from fluid to the IB. However, this approach is limited to only flexible immersed boundaries, which is why other models emerged to understand bilateral relationships between fluid and structure. For example, Goldstein *et al.* [12] developed a feedback forcing scheme to determine the externally imposed force over a fluid domain by iteratively using its velocity. Instead of the Dirac-delta function, Saiki & Biringen [13] used discrete hat function to transfer force and velocity information from IB to fluid domain and vice versa, respectively. The requirement of discrete functions or feedback forcing was alleviated by Mohd-Yusof [14] who derived direct forcing formulation and implemented it in the pseudo-spectral method. Apart from FSI, researchers also succeeded in implementing IB mechanisms using other equations. For instance, to analyse blood clotting, Fogelson & Guy [15] modelled incompressible fluid containing suspended platelets and included equations for the chemical reactions governing its stimulus response.

Significant contributions in the area of FSI came by coupling the Navier–Stokes (N-S) and Euler–Bernoulli (E-B) equations for modelling an incompressible fluid flow and an elastic IB, respectively [16–19]. N-S equations govern the nature of a fluid flow by taking care of the mass and momentum conservation. E-B theory is capable of determining small deflections of structures subjected to laterally applied loads, either pointwise or distributed over its length. This theory is based on a fourth-order differential equation that represents the relationship between the transverse displacement of a beam and the force applied to it. The main challenge in an FSI problem is how to deal with the E-B equation simultaneously with the N-S equations. The concepts on the energy transport between a beam-like structure and a Newtonian fluid, are discussed in [17]. Pontaza & Menon [18] introduced an FSI problem considering a flexible pipe inside a viscous incompressible fluid, where the flexible pipe was modelled as an E-B beam to predict its vortex-induced vibration response in the time domain.

In recent years, physics-guided machine learning (ML) approaches have earned popularity as they can exploit data-driven techniques in combination with physics-based knowledge to (i) construct descriptive models, (ii) perform computationally effective simulations, and (iii) extract useful input–output mappings or relations. For example, a deep neural network (NN) was used in [20,21] to approximate partial differential equations (PDEs), by training a large amount of data. Also, ML techniques have been explored in structural dynamics, fluid mechanics and FSI problems. In [22], data-driven recurrent NN and multi-layer perceptron were augmented with the existing domain knowledge to improve structural dynamics simulations. For fluid domains with complex boundary conditions, [23] designed an end-to-end hybrid network, V2P-Net, to infer pressure from the observed velocity fields. Moreover, a novel hydroelastic reduced order FSI model using ML can circumvent potential instability associated with the conventional Galerkin Projection method [24]. Motivated by this, in the current work, we seek for an ML-based technique to drastically reduce the computational cost involved in the traditional FSI analysis.

The present study includes fully coupled FSI by taking Peskin's IB method into consideration, which adopts an intertwined Eulerian–Lagrangian framework where the fluid domain is realized numerically

through finite differences and the immersed structure is analysed using the finite-element method (FEM). According to the conventional IB method, the velocity information is transferred from Eulerian mesh grid points to Lagrangian elements, whereas the force information is transferred just in the opposite order by using the translational property of a Dirac-delta function. The following highlights our contributions in the paper.

— The computational processes involved in the existing FSI methods are cumbersome and time-consuming. The current research aims at eliminating the cost and time in calculating structural deflections and forces.
— In our FSI implementation, we attempt to mimic the effects of fluid flow around an immersed structure, by inducing dissipation into the associated structural dynamics.
— We use a physics-guided ML technique to discover functional expressions that capture spatio-temporal dependencies in the forces and moments experienced by a structure immersed in a fluid.

In this context, we employ symbolic regression (SR) to extract explicit functional forms representing the underlying input–output relations in a given FSI data. An NN-based approximation relies on the combination of numerical weights and activations, whereas an SR-based approximation provides physically intuitive expression(s) by means of symbolic functions and operators [25]. SR was proved to be efficient in discovering numerous (100) equations from the Feynman Lectures on Physics [25]. Furthermore, in applied physics and material synthesis research [26,27], SR succeeded in unravelling symbolic energy functional expressions and extracting functional relations in hopping transport phenomena from given datasets. The remainder of this paper is organized as follows. In §2, we discuss the theoretical background of fluid dynamics, structural dynamics and FSI along with the related computational aspects. Section 3 provides the flow of our work and explains the adopted methodology. Numerical results are presented in §4. Finally, §5 concludes the current work.

# 2. Mathematical modelling

In this section, we discuss the related theory and implementation of fluid dynamics, structural dynamics and FSI.

## 2.1. Fluid dynamics and computation

### 2.1.1. Navier–Stokes equations

Consider an incompressible fluid in a two-dimensional domain that is governed by the N-S equations, as shown below:

$$\rho\left(\frac{\partial \mathbf{u}(\mathbf{x}, t)}{\partial t} + \mathbf{u}(\mathbf{x}, t).\nabla\mathbf{u}(\mathbf{x}, t)\right) = -\nabla p(\mathbf{x}, t) + \mu\Delta\mathbf{u}(\mathbf{x}, t) + \mathbf{f}(\mathbf{x}, t) \tag{2.1}$$

and

$$\nabla.\mathbf{u}(\mathbf{x}, t) = 0, \tag{2.2}$$

where $\mathbf{x} = [x, y]^T$ represents the position vector and $\mathbf{u} = [u_x, u_y]^T$ is the velocity vector evolving with time $t$; $\nabla = ((\partial/\partial x)\hat{x} + (\partial/\partial y)\hat{y})$ and $\Delta = ((\partial^2/\partial x^2) + (\partial^2/\partial y^2))$. $\rho, p, \mu$ are the fluid density, pressure and kinematic viscosity, respectively. $\mathbf{f}(\mathbf{x}, t) = [f_x, f_y]^T$ is the external force applied to the fluid domain at position $\mathbf{x}$ and time $t$. Equation (2.1) takes care of the momentum conservation and equation (2.2) ensures the incompressibility of a fluid. The fluid domain is periodic in all the directions [28]. In our study, water is used as the fluid medium with its properties: dynamic viscosity $\mu = 0.798 \times 10^{-3}$ (Ns m$^{-2}$)(30°C) and density $\rho = 997$ (kg m$^{-3}$). The domain has a constant inlet velocity $v = 0.01$ (m s$^{-1}$) with an associated Reynolds number $Re = (\rho v \, d/\mu) = 187\,406$. Note that the Reynolds number changes (locally) around the beam owing to the fluid velocity variations.

### 2.1.2. Computational aspects

Various discretization techniques, such as, FEM, finite difference approximation (FDA) and Lattice Boltzmann [29], have been used to solve the above N-S equations (2.1) and (2.2). In this study, we solve an FSI problem using the IB framework, in which the FDA scheme is adopted for simulating the N-S equations [11,30,31]. For simulation purposes, we consider a periodic square fluid domain,

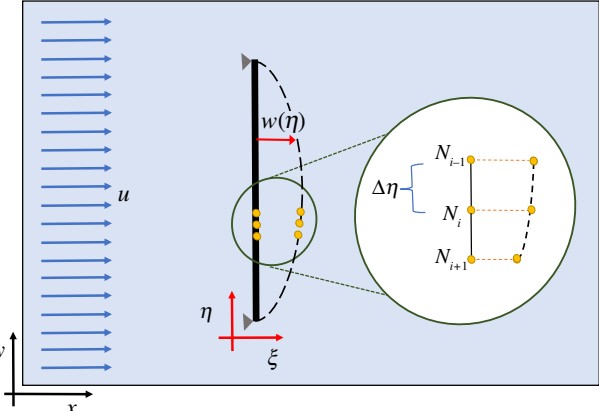

**Figure 1.** A simply supported beam immersed inside a fluid domain with the constant flow at the inlet: the beam is placed orthogonally with reference to the direction of the fluid flow; the involved coordinate systems are shown in a generic way and the nodal deflections are shown by highlighting three nodes of the beam.

$\Omega = 15\,\text{m} \times 15\,\text{m}$ and discretize it with $32 \times 20$ grid points. Let us denote the total number of discretized fluid nodes with $n$. Fluid is flowing along the length of a channel and a constant flow is maintained at its inlet. The fluid velocity is kept constant for the initial 1.5 m of the channel. However, the same varies for all the remaining grids of this channel. A simply supported beam of 10 m length is immersed in this fluid domain.

## 2.2. Beam theory and implementation

### 2.2.1. Euler–Bernoulli equation

E-B beam theory, also known as the classical beam theory [32,33], is capable of determining the bending-induced small transverse deflections of a beam. The governing equation is given by

$$\frac{\mathrm{d}^2}{\mathrm{d}\eta^2}\left(E(\eta)I(\eta)\frac{\mathrm{d}^2 w(\eta)}{\mathrm{d}\eta^2}\right) = q, \tag{2.3}$$

where $\eta \in [0, L]$, $L$ is the length of the beam and $q$ is the force per unit length distributed over the beam. Equation (2.3) depicts the relationship between the transverse displacement $w(\eta)$ of a beam and the force $q$ applied onto it, as shown in figure 1. The flexural rigidity of the beam is $(EI)$ with $E$ being the modulus of elasticity and $I$ being the second moment of area. It is worth mentioning that different orders of spatial derivatives of $w$ convey distinct attributes of an E-B beam, such as:

— $\mathrm{d}w(\eta)/\mathrm{d}\eta$ gives the slope $\theta$ at any point along the length of the beam;
— $(-E(\eta)I(\eta)(\mathrm{d}^2 w(\eta)/\mathrm{d}\eta^2))$ gives the moment in the beam; and
— $\mathrm{d}/\mathrm{d}\eta(-E(\eta)I(\eta)(\mathrm{d}^2 w(\eta)/\mathrm{d}\eta^2))$ gives the shear at beam cross sections.

In the present work, we consider a simply supported beam of length $L = 10$ m with the properties: modulus of elasticity $E = 2 \times 10^{11}\,\text{N m}^{-2}$, area moment of inertia $I = 2.133 \times 10^{-10}\,\text{m}^4$ and linear mass density $\rho_L = 8050 \times 0.008 \times 0.005\,\text{kg m}^{-1}$. The immersed beam shown in figure 1 is simply supported (hinged) at both the ends, which restricts the translational motion but allows the rotational motion of the end points. According to the associated boundary conditions, i.e. $w(0, t) = w(L, t) = 0$ and $(\partial^2 w(0, t)/\partial x^2) = (\partial^2 w(L, t)/\partial x^2) = 0$, the end nodes of the beam are stationary though the intermediate nodes get displaced owing to the fluid flow, which in turn deforms the beam without pushing (moving) it along the flow direction.

### 2.2.2. Numerical implementation

We solve the above E-B equation using FEM, for which we discretize the entire beam with $(N-1)$ number of two-nodded elements, where $N$ is the number of nodes. The beam is immersed into a fluid

of constant velocity at the inlet of the channel. The direction of fluid flow is orthogonal to the beam's longitudinal axis. While the above E-B equation is discretized using FEM, we integrate the structure with the fluid in the following way. The bending behaviour can be captured using minimum three adjacent nodes as given below [31,34]:

$$\mathbf{F}_{\text{flexural}}$$
$$= K_B \big((w_{i+1} - w_i)(\eta_i - \eta_{i-1}) - (\eta_{i+1} - \eta_i)(w_i - w_{i-1})\big) \begin{pmatrix} (\eta_i - \eta_{i-1}) + (\eta_{i+1} - \eta_i) \\ -(w_{i+1} - w_i) - (w_i - w_{i-1}) \end{pmatrix}, \qquad (2.4)$$

where $w_i := w(\eta_i)$, $\mathbf{F}_{\text{flexural}}$ is the bending force experienced by the $i$th node and $K_B$ is the bending stiffness. The axial stretching force, $\mathbf{F}_{\text{axial}}$ can be computed as

$$\mathbf{F}_{\text{axial}} = K_S \left( 1 - \frac{\Delta\eta}{\sqrt{(w_{i+1} - w_i)^2 + (\eta_{i+1} - \eta_i)^2}} \right) \begin{pmatrix} w_{i+1} - w_i \\ \eta_{i+1} - \eta_i \end{pmatrix}, \qquad (2.5)$$

where $\Delta\eta$ is the resting length between two adjacent nodes and $K_S$ is the axial stiffness; $\begin{pmatrix} w_{i-1} \\ \eta_{i-1} \end{pmatrix}$, $\begin{pmatrix} w_i \\ \eta_i \end{pmatrix}$, $\begin{pmatrix} w_{i+1} \\ \eta_{i+1} \end{pmatrix}$ are three successive nodal coordinates, which are computed via the FEM formulation.

## 2.3. How to capture fluid–structure interaction?

### 2.3.1. Peskin's immersed boundary configuration

Let $\mathbf{X}(r, t) = [X_\xi(r, t), X_\eta(r, t)]^T$ denote the deformed cartesian coordinate of the beam at $r \in [0, L]$, and $\mathbf{F}(r, t) = [F_\xi(r, t), F_\eta(r, t)]^T$ is the corresponding force vector, which is the elastic deformation force per unit area exerted by the beam on the surrounding fluid. According to Peskin's IB method, the backbone of the FSI dynamics lies in the local force and velocity transfers, governed by

$$\mathbf{f}(\mathbf{x}, t) = \int_0^L \mathbf{F}(r, t) \delta(\mathbf{x} - \mathbf{X}(r, t)) \, \mathrm{d}r \qquad (2.6)$$

and

$$\mathbf{U}(\mathbf{X}(r, t), t) = \frac{\partial \mathbf{X}(r, t)}{\partial t} = \int_\Omega \mathbf{u}(\mathbf{x}, t) \delta(\mathbf{x} - \mathbf{X}(r, t)) \, \mathrm{d}\mathbf{x}. \qquad (2.7)$$

Equation (2.6) indicates the transfer of force from structure to fluid, which is accomplished through the numerical approximation of the Dirac-delta function. On the other hand, equation (2.7) applies no-slip condition to transfer the local velocity from fluid to structure. The related theory is available in [11], and the corresponding *IB2d* method is elaborated in [31,35,36].

The velocity transferred from fluid to beam causes the whole beam's deflection. This is how the beam deflections are calculated according to the conventional detailed FSI analysis provided in [11,31].

### 2.3.2. Mimicking the fluid effect by a dissipative model

Solving the FSI problem is computationally intensive and hence time-consuming, which may be preventive for designing purposes. To alleviate the computational burden, we recall the celebrated Langevin dynamics [37,38], which captures the effect of the surrounding degrees-of-freedom (dof) in terms of dissipation and noise. Similarly, here we drop the fluid dofs and introduce a dissipative dynamics for the structure to capture the effect of fluid on it, as given in equation (2.8). We have not considered any noise term with the assumption that the deterministic force is large enough to allow us to consider that the noise term is negligible:

$$C_V \frac{\partial w(\eta, t)}{\partial t} + \frac{\partial^2}{\partial \eta^2} \left( E(\eta) I(\eta) \frac{\partial^2 w(\eta, t)}{\partial \eta^2} \right) = q(\eta) \qquad (2.8)$$

and

$$C\dot{\boldsymbol{\psi}} + K_G \boldsymbol{\psi} = \mathcal{F}. \qquad (2.9)$$

The PDE (2.8) represents a dynamic E-B beam with dissipation [39]. The first term on the left-hand side of

this equation stands for the damping force with a damping coefficient $C_V$. $q$ is an external force on the right-hand side of the same. Upon finite-element (FE) discretization, it takes the form given in (2.9), where, $C$, and $K_G \in \Re^{2N \times 2N}$ are the damping coefficient matrix and the global stiffness matrix, respectively. Here, $\psi \in \Re^{2N \times 1}$ is the displacement (linear and angular) and $\dot{\psi}$ is the discretized velocity vector. The force vector $\mathcal{F} \in \Re^{2N \times 1}$ stands for the generalized force comprised of both the nodal force and moment.

To characterize the dissipation, the Rayleigh damping model is used here, which represents the damping as a linear combination of the stiffness and mass matrices, given as

$$C = \mu M_G + \lambda K_G, \tag{2.10}$$

where $\mu$, $\lambda \in \Re^1$ are the constants chosen as one and $M_G \in \Re^{2N \times 2N}$ is the mass matrix. We can decouple the dynamics by inverting $C$ such that the modified force vector and the stiffness matrix in $\mathcal{F}$ become: $\tilde{\mathcal{F}} = C^{-1}\mathcal{F}$ and $\tilde{K}_G = C^{-1}K_G$. This inverse operation decouples the forces applied in different dof.

We feed the deflection, calculated using the conventional way [11], into the proposed dissipative model (2.8) to generate the nodal forces/moments for the ML analysis. The next section states the objective, describes the flow of the processes involved in producing the required data, and calls for an adept ML technique to extract the functional mapping of our main concern.

# 3. Formulation and methodology

The *auxiliary goal* of this work is to develop an indirect yet effective way of computing forces and moments experienced by an immersed simply supported beam, by incorporating dissipation into the associated structural dynamics. The aim is to generate nodal forces/moments from the deflections calculated by the conventional FSI analysis. The *primary goal* is to capture the functional relationship(s) between the computed forces/moments and the spatio-temporal variables, to alleviate the extensive costs involved with numerically solving nonlinear PDEs. The reconstructed forces/moments in continuum functional forms are expected to produce the same deflections as obtained by the conventional FSI analysis.

## 3.1. Connecting processes and data curation

Partitioned solution procedure [40] is employed in solving the underlying IB problem in FSI. The simulation is run for $t_f = 15$ s with an increment of $\Delta t = 5 \times 10^{-5}$ s, and the corresponding discrete time step is denoted by $k \in \{1, 2, \ldots, (t_f/\Delta t)\}$. The following steps are involved in the simulation process.

**Flow of detailed FSI analysis:**
*for* $k = 1, 2, \ldots, (t_f/\Delta t)$

  (i) numerically solve the N-S equations, (2.1) and (2.2) on the fluid domain to obtain the updated velocity $\mathbf{u}_j^{k+1}$ from $\mathbf{u}_j^k$ and $\mathbf{f}_j^k$, where subscript $j$ denotes the respective field variable computed at the $j$th node (after discretization);
 (ii) the updated velocity of the fluid nodes is transferred to the beam as per equation (2.7), which updates its nodal locations by: $\mathbf{X}_i^{k+1} = \mathbf{X}_i^k + \mathbf{U}_i^{k+1}\Delta t$. Here, $i$ denotes the $i$th node of the beam after discretization;
(iii) the updated nodal position $\mathbf{X}_i^{k+1}$ of the beam is used in equations (2.4) and (2.5) to calculate $\mathbf{F}_i^{k+1}$;
(iv) the beam deformation force $\mathbf{F}_i^{k+1}$ is then transferred to the fluid grid points in the form of fluid force $\mathbf{f}_j^{k+1}$, according to equation (2.6).

  *end*

By following the above procedure, we calculate the nodal displacements $\mathbf{X}_i$. Next, the angular displacements induced in these coordinates are extracted for all $i$ and denoted by $\theta_i$. By using $\psi_i$ and $\dot{\psi}_i$ in equation (2.9), we compute the generalized nodal forces $\tilde{\mathcal{F}}_i = [F_i, M_i]^T$. Note that our proposed dissipative model (2.8) provides decoupled forces/moments, the ground truth data for ML, from the deflections obtained by the *IB2d* method [11,31,35,36]. The dataset fed to the employed ML technique, is: $\{\{t, (x_i/L), F_i, M_i\}_{i=1}^N\}_{\bar{k}=1}^{t_f/\Delta t}$; $\bar{k} = \{1, 101, 201, \ldots, (t_f/\Delta t)\}$. Figure 2 gives a visualization of the problem formulation along with the proposed methodology. To enhance clarity, we provide a qualitative comparison between the traditional approach and our ML-based approach, as depicted in

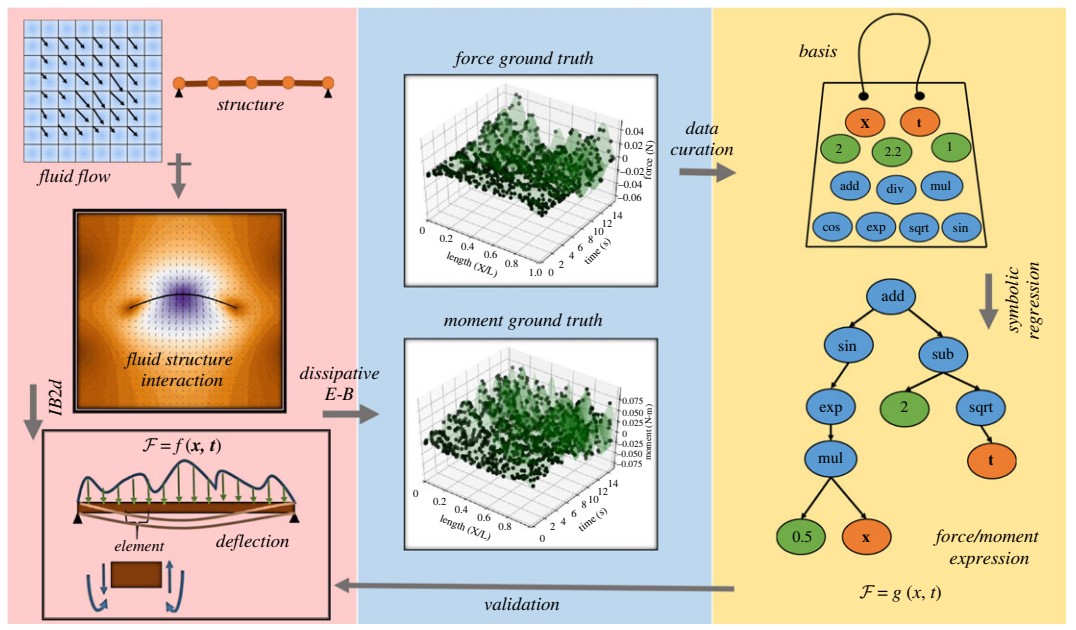

**Figure 2.** A flow diagram of the problem formulation and proposed methodology: (*a*) the processes involved in FSI interaction dynamics, (*b*) the dataset, and (*c*) depicts the ML solution.

**Table 1.** A qualitative comparison between the existing and our proposed FSI analysis approaches to highlight some of the key factors and advantages.

| attributes | traditional FSI analysis | our ML-based approach |
|---|---|---|
| Navier–Stokes solution handling | explicitly solves Navier–Stokes equations governing fluid motions, which involves huge computational overhead | bypasses solving Navier–Stokes by mimicking fluid flow effects with proper forcing functions into dissipative Euler–Bernoulli beam model |
| structural displacement calculation | displacements are calculated via numerical integration involving force and velocity transfers from structure and fluid and vice versa | displacements are calculated by feeding SR-generated forces and moments into the dissipative Euler–Bernoulli beam model |
| cost of computation | computationally expensive owing to simultaneously solving numerical integration | computationally inexpensive owing to elegant functional forms of forces/moments |

**Table 2.** Parameters used in the symbolic regression.

| fit | population_size | tournament_size | parsimony_coefficient | p_crossover | p_mutation |
| --- | --- | --- | --- | --- | --- |
| force | 20 000 | 110 | 0.003 | 0.7 | 0.29 |
| moment | 18 900 | 100 | 0.002 | 0.7 | 0.29 |

table 1. The following illustrates the currently employed ML technique to extract functional relations from force/moment ground truth data.

## 3.2. Machine learning for extracting functional forms

We call for an adept ML technique, SR, to extract functional relations between the input–output variables contained in the data $\{\{t, (x_i/L), F_i, M_i\}_{i=1}^{N}\}_{\bar{k}=1}^{t_f/\Delta t}$, generated from our FSI simulations. Here, the input variables are: time $t_{\bar{k}}$ and space $\{x_i/L\}$, and the output variables are: nodal force $\{F_i\}$ and moment $\{M_i\}$ $\forall i$ at all the saved time-instants.

### 3.2.1. Symbolic regression

A symbolic input–output mapping offers interpretability of the underlying physics, which is the motivation behind using SR over other ML techniques [25]. Here, we provide a precise mathematical overview of a typical SR problem. Consider a dataset $(\chi, y)$ comprised of $n$ input and one output variables, where $\chi \in \mathfrak{R}^n, y \in \mathfrak{R}^1$. SR seeks to find the mathematical expression of a function $g(\chi): \mathfrak{R}^n \to \mathfrak{R}^1$ that minimizes an error functional $\mathbb{E}_m(y, g(\chi))$ between the actual and predicted values, where $\mathbb{E}_m$ stands for an error metric which can be mean absolute error or mean square error or other reasonable customized errors [41]. The solution to this functional optimization problem is the desired optimal function, given as: $g^*(\chi) = argmin_g \mathbb{E}_m(y, g(\chi))$.

### 3.2.2. Genetic programming

SR uses genetic programming (GP) to determine the expression that best captures the input–output relationship from a given data. GP is an evolutionary algorithm used to solve functional optimization problems, where candidate solutions are represented by trees made up of variables, numbers, symbols and functions [41,42]. These candidate trees evolve with generations using *tournament selection* followed by *crossover* and *mutation* operations [42]. In tournament selection, random candidates are selected from a population to construct a pool to be engaged in a tournament. The tournament winner, i.e. the tree with the highest fitness (minimum regression error), is carried forward to the next operation. It is worth noting that a large pool size discourages weak individuals to participate in the tournament. The exploration and exploitation phases of the evolutionary search are taken into account by mutation and crossover operations, respectively, as described below.

GP applies crossover and mutation operators within the candidate tree structures. A crossover propagates useful information from parents to offspring to improve the fitness of the next generation. Crossover takes the winner of a tournament and selects a random sub-tree from it to be replaced by a donor coming from another tournament; therefore, this operation requires at least two prior tournaments to be over to find out a parent and a donor [42,43]. As per the consequence of the sub-tree interchange in a crossover operation, also, the donor has a randomly selected sub-tree that is inserted into the parent to form an offspring. On the other hand, the mutation has the potential to incorporate extinct functions into the population to maintain diversity, and thus it enhances the exploration capability of the evolution process. Mutation can be of different types, such as: sub-tree mutation, hoist mutation and point mutation, which can amend the candidate tree structures by random replacement(s) of sub-trees and/or nodes [41]. For further details and visualization, please see appendix A.

# 4. Numerical results

In this section, we present the force and moment approximations achieved by a physics-guided ML approach, and further validate these solutions on the beam deflections.

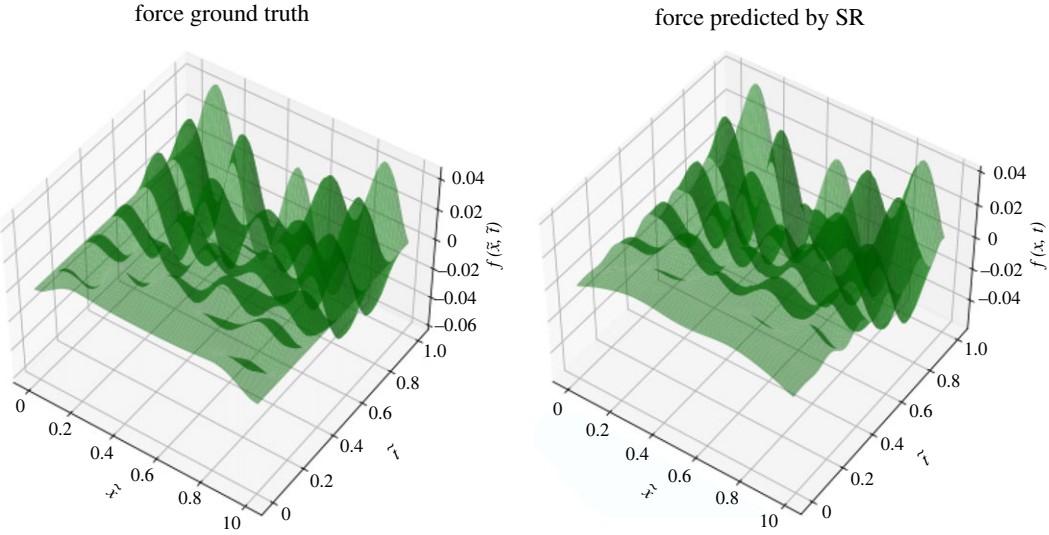

**Figure 3.** Symbolic regression (SR) outcome fitted to the ground truth data of nodal forces with respect to the spatio-temporal independent variables. The independent axes are non-dimensionalized by: $\tilde{x} = x/L$ and $\tilde{t} = t/15$. The corresponding fitting error after 250 generations is: MAE=0.00351.

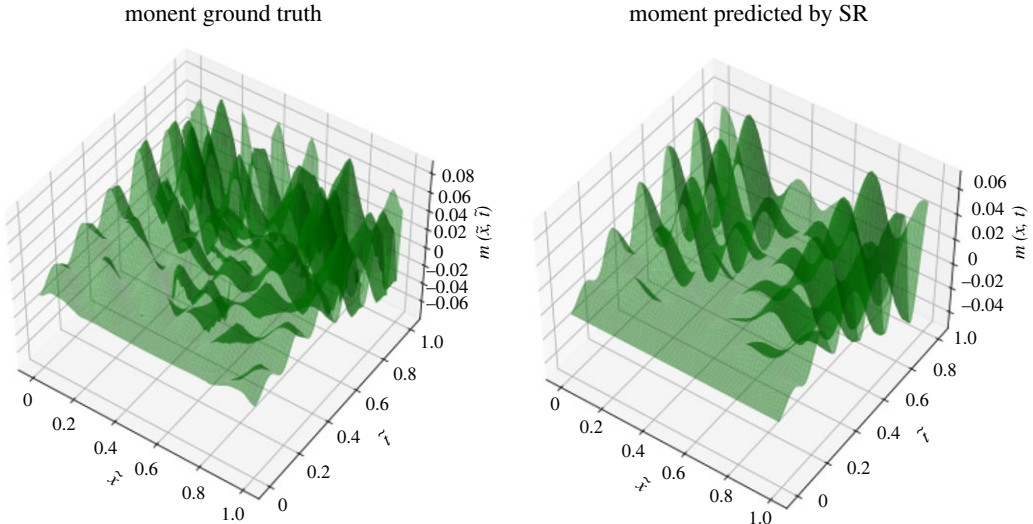

**Figure 4.** Symbolic regression (SR) outcome fitted to the ground truth data of nodal moments with respect to the spatio-temporal independent variables. The independent axes are non-dimensionalized by: $\tilde{x} = x/L$ and $\tilde{t} = t/15$. The corresponding fitting error after 250 generations is: MAE = 0.00874.

## 4.1. Functional approximations by symbolic regression

### 4.1.1. Machine learning implementation

Our dataset contains the force and the moment values for 100 time-stamps while considering 65 beam nodes for each time-stamp. The associated independent variables are: $a \in [0, 1] = \eta/L$ denoting the node locations per unit length of the beam and $b \in [0, 15] = t$ representing time in seconds. The output variables are the nodal forces at all the saved time-stamps, $t_{\bar{k}}$. In case there exists slight asymmetry in the force/moment data, then we make it fully symmetric prior to feeding it to ML. We employ the '*gplearn*' solver [41] to implement SR for extracting input–output symbolic relations. The set of basis functions used in force fitting is: {*add, sub, mul, div, cos, sqrt, exp*}, where 'div', 'sqrt' and 'exp' are protected functions to avoid numerical overflow errors [41]. In the case of moment fitting, one additional function 'sin' is included in the basis set. Intuitively, we choose 'sin', 'cos' and 'exp' in

the basis set owing to the periodic behaviour and the decaying nature observed in the force/moment ground truth landscapes. The error metric used in the regression is *mean absolute error* (MAE). The tuning parameters involved in SR are listed in table 2. Here, the total mutation parameter, p_mutation, is distributed into three parts: p_subtree_mutation=0.11, p_hoist_mutation=0.07 and p_point_mutation=0.11. The compactness of an SR outcome expression depends on the corresponding parsimony coefficient. The hyper-parameter values of the population_size, tournament_size and parsimony_coefficient are tuned using a grid search mechanism for accurate fitting. Figures 3 and 4 exhibit the force and moment fitting performances achieved by SR in 250 generations, respectively.

### 4.1.2. Force approximation

The continuous approximation of the spatio-temporal force experienced by a simple supported steel beam of size: (10 m × 0.005 m × 0.008 m) immersed in water with speed 0.01 m s$^{-1}$ at the channel inlet, is as follows:

$$
\begin{aligned}
\hat{f}(a, b) = &(-b^{0.5}(-ab + (-2b + \exp(\exp(0.449\exp(a))) \\
&- \exp(\cos(1.418 + \cos(a)/(a + 0.117)))/a)^{0.5}) \times \cos(0.032b(a - 0.619b) - 2b \\
&+ (b^{0.5} \times (a - 0.169) + b)^{0.5} - \exp(a) + \exp(\exp(a^{0.5}))) - 2b \\
&- (a - b) \times \cos(0.032b \times (-ab + \cos(b - (1.418 + 1.191/a) \times (a - 0.169)))) - 2b \\
&+ (a - 0.389) \times (b + \cos(0.011b^2)) - \exp(a) + \exp(\exp(a^{0.5}))) \\
&- (-b((a - 0.169) \times \exp(a^{0.5}) - \cos(\cos(\exp(0.094b) - \exp(\cos(a - 0.8)))))^{0.5} + \cos(ab)) \\
&\times \exp((b^{0.5} \times (a - 0.169)^2 - \cos(a))^{0.5}) \\
&\times \cos(a^{0.5} - 0.032b(-0.032ab(-ab - \cos(b) - 2\cos(0.183b \times \exp(a)) - 0.032) - ab) + 2b \\
&- \exp(\exp(\cos(a)))^{0.5} - \exp(\exp(\cos(a)))) \\
&+ \exp(\exp(\cos(0.609 \times \exp(-b)))) + \exp(\cos(0.032ab^2))) \\
&\times \cos(2.718 \times (a - 0.169) - \cos(\cos((1.418 + 1.191/(a + 0.117)) \times (a - 0.169)))).
\end{aligned}
\tag{4.1}
$$

### 4.1.3. Moment approximation

The continuous approximation of the spatio-temporal moment experienced by a simple supported steel beam of size: (10 m × 0.005 m × 0.008 m) immersed in water with speed 0.01 m s$^{-1}$ at the channel inlet, is as follows:

$$
\begin{aligned}
\hat{m}(a, b) = &(a - 0.491)^{0.5} \times (b\exp(\cos(a))^{0.5} \times \sin(11.905a + 2b + (b^{0.5} + b - 2.630)^{0.5} \\
&- \exp(\sin(a - 0.412)^{0.5}) + 0.684) - b \times \sin(11.905a - b^{0.5} - 2b + 1.642) \\
&+ (0.144b(b + 2.630) \times (-\sin(23.809a - b^{0.5} - 2b + 4.576) \\
&+ \sin(4.073 \times a^{0.5} - 23.809a + b^{0.5} + 2b - \sin((b - 0.967)^{0.5}) - 1.946) + 0.684) \\
&+ b \times ((-10.905a + 2b)^{0.5} - 0.967) \times \sin(23.809a + 2b + 2(-b^{0.5} + b + 2.630)^{0.5} \\
&- 0.662\exp(a) + 0.684) + 2b + 0.380b^{1.5} \\
&\times (0.684 - \sin(\sin(11.905a - b^{0.5} - 2b + 1.642))) + (3b + 0.967) \\
&\times \sin(\sin(b^{0.5}))) \times \cos(0.372b^{0.5}) - 0.191)
\end{aligned}
\tag{4.2}
$$

In figure 3, the force fitting performance is incredibly good wherein the predicted (approximated) force accurately captures most of the peaks and valleys present in the ground truth. The moment fitting performance in figure 4 is descent, and the moment expression comes out to be more compact than that of the force. The outcome expressions (4.1) and (4.2) are lengthy true, however, their evaluations are much easier than computationally expensive finite-element codes. Let us examine what can be inferred from these equations. To estimate the complexity of the ground truth, SR solutions (4.1) and (4.2) incorporate various function combinations like sin ($e^a$), $e^{\sin(a)}$, $e^{e^{\sin(a)}}$. The interpretability of these elementary combinations extracted by SR, is given below.

— **sin(a)** versus **e$^{\sin(a)}$**: the function $e^{\sin(a)} \in [e^{-1}, e^1]$ is always positive and has a different range of values than sin ($a$) $\in [-1, 1]$, although their periodicities do not differ;

**Table 3.** Mean, standard deviation (s.d.) and maximum values of the difference vector between the reference and ML outcome deflection profiles w.r.t time, for different number of beam nodes. (Note: a consistent fluid domain grid resolution of $(32 \times 20)$ is used for different discretizations of the beam.)

| time (s) | difference (%L) | $N = 65$ | $N = 100$ | $N = 200$ |
|---|---|---|---|---|
| 1.5 | mean | 0.013 | 0.013 | 0.014 |
| | s.d. | 0.008 | 0.012 | 0.014 |
| | max | 0.030 | 0.036 | 0.042 |
| 3 | mean | 0.042 | 0.053 | 0.067 |
| | s.d. | 0.033 | 0.039 | 0.045 |
| | max | 0.097 | 0.117 | 0.141 |
| 4.5 | mean | 0.018 | 0.033 | 0.054 |
| | s.d. | 0.016 | 0.015 | 0.022 |
| | max | 0.054 | 0.060 | 0.084 |
| 6 | mean | 0.027 | 0.042 | 0.070 |
| | s.d. | 0.020 | 0.022 | 0.033 |
| | max | 0.066 | 0.074 | 0.121 |
| 7.5 | mean | 0.027 | 0.034 | 0.067 |
| | s.d. | 0.023 | 0.027 | 0.043 |
| | max | 0.084 | 0.087 | 0.139 |
| 9 | mean | 0.016 | 0.030 | 0.067 |
| | s.d. | 0.010 | 0.024 | 0.038 |
| | max | 0.035 | 0.070 | 0.129 |
| 10.5 | mean | 0.025 | 0.030 | 0.055 |
| | s.d. | 0.016 | 0.023 | 0.027 |
| | max | 0.052 | 0.066 | 0.110 |
| 12 | mean | 0.035 | 0.055 | 0.085 |
| | s.d. | 0.023 | 0.034 | 0.056 |
| | max | 0.073 | 0.111 | 0.167 |
| 13.5 | mean | 0.056 | 0.082 | 0.109 |
| | s.d. | 0.051 | 0.057 | 0.041 |
| | max | 0.172 | 0.210 | 0.202 |
| 15 | mean | 0.068 | 0.092 | 0.109 |
| | s.d. | 0.059 | 0.073 | 0.069 |
| | max | 0.164 | 0.202 | 0.190 |

— **sin(a)** versus **sin(eᵃ)**: the range of values of $\sin(e^a) \in [-1, 1]$ is same as that of $\sin(a)$; however, the frequency of the former function increases exponentially though the latter has a constant frequency;

— **eᵃ** versus **sin(eᵃ)** & **e**$^{\sin(a)}$: the function $e^a \in (0, \infty)$ is non-periodic and unbounded, whereas **sin(eᵃ)** & **e**$^{\sin(a)}$ are periodic and bounded functions.

Thus, equations (4.1) and (4.2) unfold the periodic behaviour in combination with exponential growth or decay, describing the nature of the underlying force/moment variations in space and time.

The moment is applied onto the rotational dof, whereas the force is applied onto the translational dof. As the rotations are more undulating than the displacements, intuitively, we expect the moment landscape to be more irregular than the force landscape with several peaks and valleys. Therefore, the currently employed ML technique, i.e. SR, confronts challenges to capture the spatio-temporal

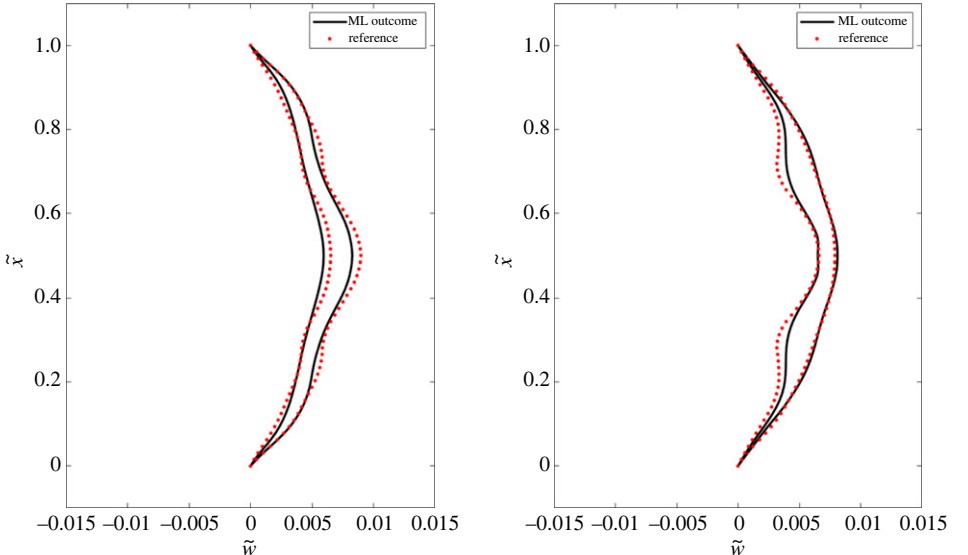

**Figure 5.** A visual comparison between the beam deflection profile simulated by detailed FSI analysis (dotted red) and by using ML (solid black): the first diagram shows the deflections at $t1$ and $t2 = 4$ s and 7 s along the fluid flow, and the second one shows the deflections at $t3$ and $t4 = 9.25$ s and 12.25 s against the fluid flow. Here, the independent and dependent axes are non-dimensionalized by $\tilde{x} = x/L$ and $\tilde{w} = w/L$.

**Table 4.** A performance evaluation comparison of different fluid domain grid resolutions, carried out with reference to the deflection simulated using higher resolved grids: 100 grids along $x$-axis $\times$ 100 grids along $y$-axis. The root mean square error (RMSE) is calculated between the deflections ($\in \mathfrak{R}^{65 \times 3000}$) obtained with different grid resolutions, at the corresponding Lagrangian points (65) for the same time instances (3000).

| grid resolution | time consumption (min) | simulation error (RMSE) |
|---|---|---|
| 100 grids along $x$-axis $\times$ 100 grids along $y$-axis | 70 | 0.00 (ref) |
| 32 grids along $x$-axis $\times$ 20 grids along $y$-axis | 14 | 0.0189 |
| 20 grids along $x$-axis $\times$ 32 grids along $y$-axis | 14 | 0.0293 |
| 32 grids along $x$-axis $\times$ 32 grids along $y$-axis | 17 | 0.0310 |

functional relations in the moment profile. To tackle this high complexity, we tried to empower SR by feeding it more basis (elementary composition) functions. Still, the accuracy of the achieved moment fit is slightly worse than that of the force fit; however, the reported result is the best among various solutions obtained with different parameter settings in SR.

## 4.2. Validating beam deflections

The continuous expressions in equations (4.1) and (4.2) enable us to evaluate force and moment at any point on the beam. To eliminate slight asymmetries present in the force approximation (4.1), we apply the following operation onto it:

$$\hat{F}(a, b) = \frac{\hat{f}(a, b) + \hat{f}(1 - a, b)}{2}.$$

In similar fashion, we also obtain a fully symmetric moment approximation $\hat{M}(a, b)$. The fully symmetric forces and moments, $\tilde{\mathcal{F}} = [\hat{F}, \hat{M}]^T$, are fed into equation (2.9) to compute $\hat{\boldsymbol{\psi}}$, which is the ML estimate for $\boldsymbol{\psi}$. In order to validate the interpolation ability of the achieved functional forms, we calculate beam deflections for varying number of nodes and present a performance evaluation study in table 3. In this context, note that for the detailed FSI analysis, Lagrangian discretization (ds) is taken: 0.5×Eulerian discretization (dx), as considered in IB2d [11]. For $N = 65$, figure 5 shows a visual comparison between the beam deflections obtained by detailed FSI analysis and by using SR. The

slight differences between the reference and ML outcome in figure 5, indicate that the reconstructed forces capture the low-order modes better than the high-order modes of oscillations. Please see apendix A for additional simulation results.

# 5. Conclusion

In this study, we have devised an ML-based strategy to solve an FSI problem by extracting the underlying functional forms that capture the effect of the fluid on an immersed beam. The ML estimate of the forces, applied to the EB equation with dissipation, performs well in predicting the deflections of the immersed beam as found from the detailed FSI simulations. The functional forms (expressions) generated by the employed SR technique, accurately fits the ground truth force landscape. Moreover, the achieved expressions provide interpretability in understanding the spatio-temporal behaviour of the fluid forces/moments on the immersed beam. Although the input to SR is discrete, the results demonstrate that the output force/moment functions, continuous in space and time, can capture the beam deflections following different FE discretizations. To this end, the proposed approach helps us reduce the computational overhead involved in the traditional FSI analysis and produces continuum functional relations.

In the future, we plan to extend the study for higher dimensional FSI problems involving a two-dimensional plate or a three-dimensional object immersed in the fluid. We believe this work has potential to generate computationally tractable and interpretable solutions to real-life aerodynamic and marine engineering applications.

Data accessibility. The data and the codes for the manuscript are available in the Dryad repository [44].

Authors' contributions. T.S.: formal analysis, investigation, methodology, software, validation, writing—original draft; A.S.: data curation, investigation, methodology, software, validation, visualization, writing—original draft; R.D.: formal analysis, methodology, project administration, supervision, visualization, writing—original draft, writing—review and editing; A.D.: conceptualization, formal analysis, investigation, visualization, writing—review and editing; S.J.: funding acquisition, investigation, resources, writing–review and editing; S.S.: conceptualization, methodology, project administration, resources, supervision, writing–original draft, writing—review and editing.

All authors gave final approval for publication and agreed to be held accountable for the work performed therein.

Competing interests. We declare we have no competing interests.

Funding. R.D. and S.J. acknowledge funding from the Accelerated Materials Development for Manufacturing Program at A\*STAR via the AME Programmatic Fund by the Agency for Science, Technology and Research under grant no. A1898b0043.

# Appendix A

In this section, we provide intricate implementation details along with the associated challenges, and discuss the extrapolation capability of the currently adopted approach.

## A.1. Implementation details

### A.1.1. Fluid domain resolution

We have performed the simulation with different grid resolutions, however, the resolution ($32 \times 20$) has been considered to reduce the computational overhead without compromising the accuracy in deflections. In support of this argument, a quantitative evaluation study is presented in table 4.

### A.1.2. Stiffness calculation

In accordance with the fibre model reported in [30], the bending effect is realized by torsional springs (of stiffness $K_B$) connecting every three consecutive Lagrangian points, $(i-1)$th, $i$th and $(i+1)$th, while the axial deformation is realized by linear springs (of stiffness $K_S$) linking every two consecutive Lagrangian points, $i$th and $(i+1)$th.

The bending stiffness, $K_B$ in equation (2.4), is computed according to the following procedure. For calculating $K_B$, a set of distinct external forces $\bar{\mathbf{f}}_j$ for $j = 1, 2, \ldots, m$ are applied on the beam, which deflects all the nodes throughout it as per the following equation:

$$\bar{\mathbf{f}}_j = [K_G]\tilde{\boldsymbol{\psi}}_j. \tag{A 1}$$

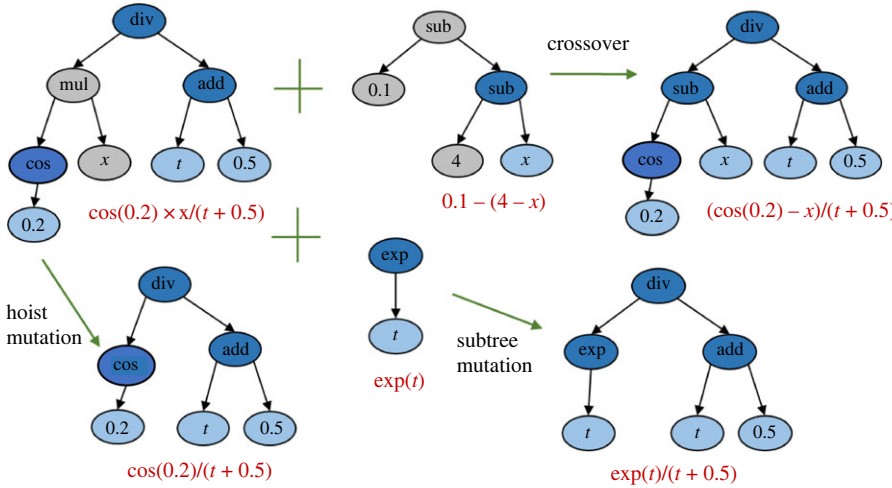

**Figure 6.** A visualization of the crossover and mutation operations involved in the evolution of candidate solution trees.

In equation (A 1), $K_G \in \mathfrak{R}^{2N \times 2N}$ is the global stiffness matrix; $\tilde{\boldsymbol{\psi}}_j \in \mathfrak{R}^{2N \times 1}$ represents the displacement vector for all nodes, which is sparse everywhere except at the node onto which the force $\bar{\mathbf{f}}_j \in \mathfrak{R}^{2N \times 1}$ has been applied. The lateral displacement of the $i$th node is denoted by $\tilde{w}_{i,j}$ and is contained within $\tilde{\boldsymbol{\psi}}_{i,j}$. The bending stiffness is computed by using the slope of the force versus nodal displacement. These stiffness values for all the nodes, $i = 1, 2, \ldots, N$, are calculated according to the FDA, as follows:

$$K_{B_{ij}} = \frac{\bar{f}_{i,j+1} - \bar{f}_{i,j}}{\tilde{w}_{i,j+1} - \tilde{w}_{i,j}}. \tag{A 2}$$

The axial stiffness, $K_S$ in equation (2.5), is computed directly as

$$K_S = \frac{(N-1)EA}{L}. \tag{A 3}$$

The bending stiffness and axial stiffness of the beam increase as the number of discrete nodes increases. The axial stiffness increases at a linear rate, and the bending (flexural) stiffness increases at a nonlinear rate.

### A.1.3. Machine learning parameter tuning

Parameter tuning is an important aspect of SR. SR relies on GP, where a balance between exploration and exploitation needs to be maintained for optimal performance. To understand the choice of tuning parameters, first, we need to realize their roles and purposes. Here, $p$-crossover parameter takes care of the exploitation and $p$-mutation $= (p$-subtree-mutation $+ p$-hoist-mutation $+ p$-point-mutation$)$ takes care of the exploration of the search space. $p$-subtree-mutation is an aggressive mutation parameter since it can replace significant genetic material, i.e. a subtree composed of nodes and branches, from the tournament winners to introduce extinct functions/operators into a population of candidate solutions; $p$-hoist-mutation is a bloat-resisting mutation parameter that removes genetic material by hoisting a random subtree into the original subtree's location; and $p$-point-mutation is a nominal mutation parameter that replaces random nodes from the tournament winners to introduce new function/operator [41]. Figure 6 gives a visualization of the crossover and mutation operations.

### A.1.4. Reproducibility of results

In the currently employed SR solver, i.e. gplearn, random numbers are used at two places. The first random number is used to apply a permutation for shuffling the input data points to the solver. The second random number is used in the GP for candidate evolution. The SR-generated forces/moments are reproducible given that the random numbers are called from a fixed seed during its *python* implementation.

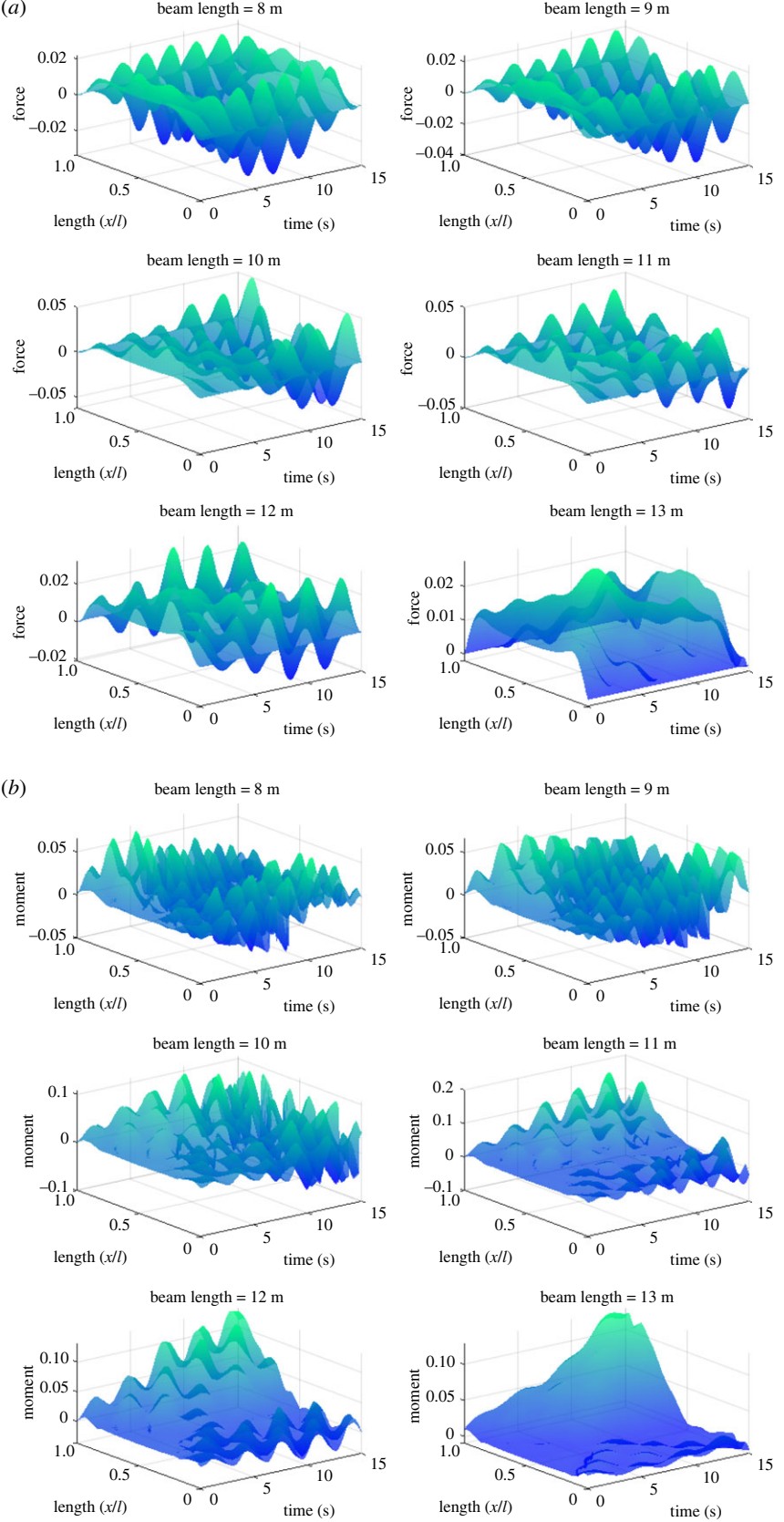

**Figure 7.** Force and moment landscapes for different beam lengths. (*a*) Various force distributions for different lengths of beams while all other parameters are constant. The spatio-temporal patterns are similar for 9, 10, 11, 12 m beams, however, they differ significantly for 8 and 13 m beams. (*b*) Various moment distributions for different lengths of beams while all other parameters are constant. The spatio-temporal patterns are similar for 9, 10 m beams and 11, 12 m beams, however, they differ significantly for 8 m and 13 m beams.

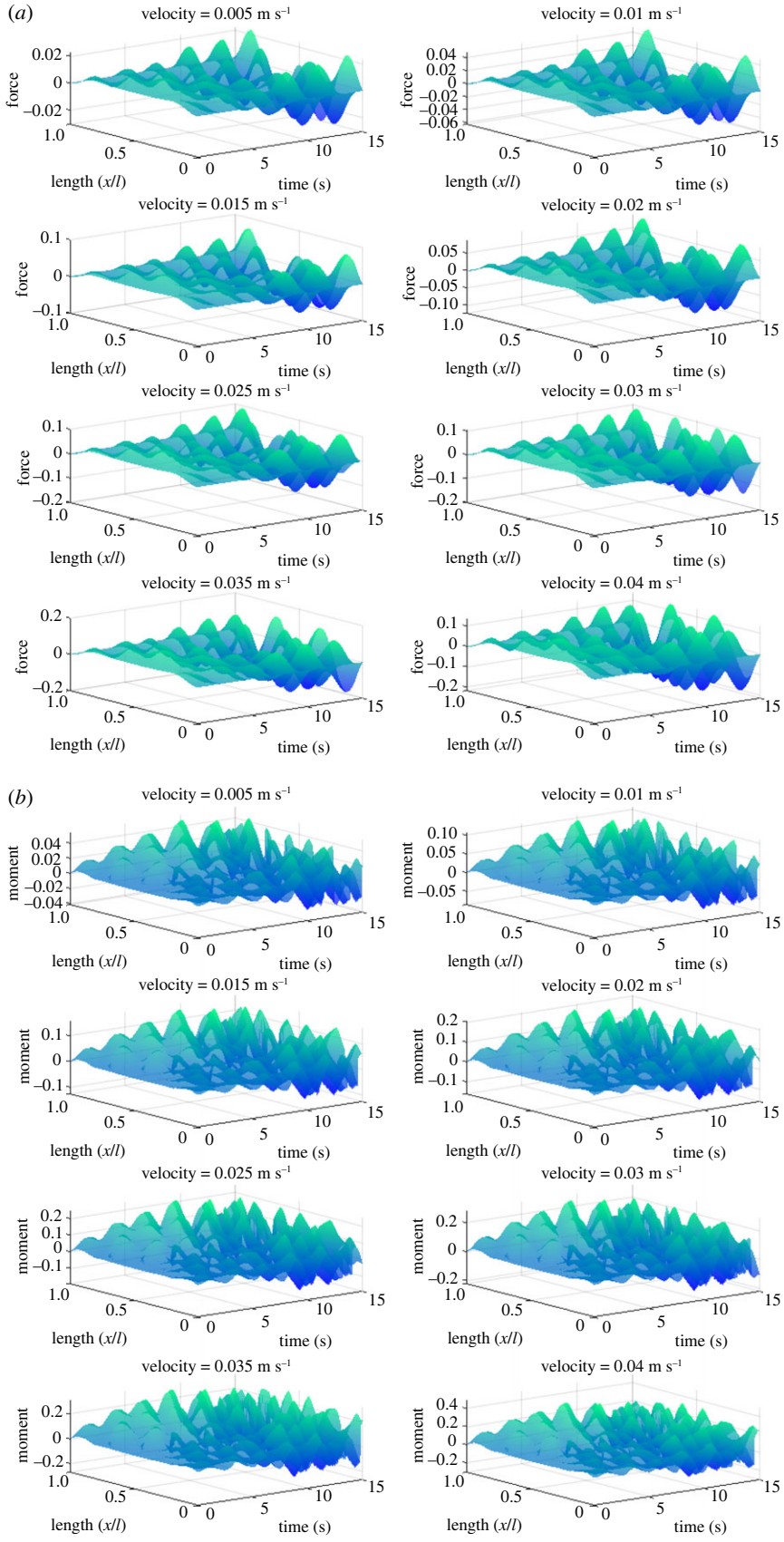

**Figure 8.** Force and moment landscapes under different fluid velocities. (*a*) Different force distributions under various fluid velocities while all other parameters are constant. The spatio-temporal patterns are very similar under velocities 0.005–0.02 m s$^{-1}$, however, a few peaks start growing when velocity >0.02 m s$^{-1}$. (*b*) Different force distributions under various fluid velocities while all other parameters are constant. The spatio-temporal patterns are very similar under velocities 0.005–0.04 m s$^{-1}$.

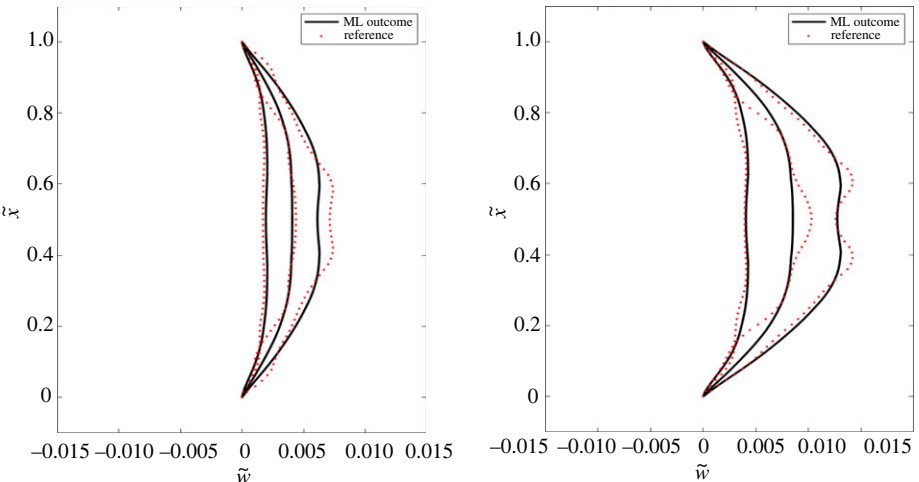

**Figure 9.** Extrapolation capability of the ML-generated expressions, obtained for a 10 m beam immersed into a fluid flow of inlet velocity 0.01 m s$^{-1}$, in reconstructing beam deflections under different conditions. Here, the independent and dependent axes are non-dimensionalized by $\tilde{x} = x/L$ and $\tilde{w} = w/L$. (*a*) Reconstructed deflections of a beam of length 11 m. (*b*) Deflection reconstructed for an inlet velocity of 0.02 m s$^{-1}$.

### A.1.5. Implementation challenges

The currently employed ML technique, i.e. SR, succeeds in capturing the spatio-temporal functional relations present in the ground truth forces/moments profiles. However, during the implementation, we confronted challenges as mentioned below:

— SR relies on GP, which is very sensitive to the input parameters like 'population size' and 'tournament size'. Hence, we had to perform a grid search for proper parameter tuning;
— the choice of basis (elementary composition) functions is critical and important. In practice, we conducted many trial and error simulation runs before coming up with a suitable library of basis functions. The ground truth moment profile had more complexity than the force profile. To tackle this, we empowered SR by feeding it a slightly bigger library of basis functions; and
— to avoid numerical overflow errors [41], we protected some of the basis functions like 'sqrt', 'exp' in GP. This minute step is important in simulations.

In future, our proposed framework can be extended from two-dimensional to three-dimensional problems while carefully addressing two major factors, i.e. parameter sensitivity and the choice of basis functions, during the implementations.

### A.2. Extrapolation to generality

The employed ML technique, SR, fits continuous symbolic functions passing through the given data points. Thus, the achieved outcome expressions have adequate interpolation capability, as supported by the performance evaluation study where the same functional forms of forces/ moments work under different FE discretizations in table 2. We are able to extrapolate the ML solutions to some extent, however, the extrapolation ability of these expressions still remains a concern. The current focus is on capturing the spatio-temporal patterns within a small scale. In future, we would like to extend our approach to a large scale while neglecting high-resolution spatio-temporal oscillations.

However, it is challenging for the ML-based solver to fit extrapolated data owing to two factors: (i) the innate limitation of the fitting approach by SR, and (ii) the transient behaviour of the vibration response. In case the length of the beam is increased while other parameters remain constant, then the extrapolation can work within a small range, as justified by figure 7*a*,*b*. In figure 7*a*, the distribution (spatio-temporal patterns) of force do not change much for beam lengths 9, 10, 11, 12 m, and in figure 7*b*, the distribution of moment do not change much for beam lengths 9, 10 m and for 11, 12 m; only their magnitudes change to some extent. So, the same force and moment functional forms generated by SR for a 10 m beam can extrapolate to 9–12 m beams in combination with proper scaling factors. Furthermore, the spatio-

temporal patterns of the force and moment landscapes remain similar when the fluid velocity (at the inlet) is varied within a range of 0.005–0.04 m s$^{-1}$ for a 10 m beam, as depicted in figure 8$a$,$b$. The force and moment patterns are very close for a velocity range of 0.005–0.04 m s$^{-1}$ except a few peaks in the forces start growing when velocity is more than 0.02 m s$^{-1}$. Thus, the extrapolation is possible for a range of velocities as well. Figure 9 shows that the force and moment expressions obtained for a 10 m beam, along with the respective scaling factors (1.07, 0.91), can reconstruct the deflections of a 11 m beam for 5 s. Figure 9 also shows that the force and moment expressions obtained for a 10 m beam immersed into a fluid flow of inlet velocity 0.01 m s$^{-1}$, along with the respective scaling factors (2, 2), can reconstruct the deflections under different inlet velocity 0.02 m s$^{-1}$ for 5 s. Nevertheless, the extrapolation ability along the time axis is challenging, which we plan to tackle in future by feeding data that spans over a long time.

Unlike NN-based ML techniques, the currently employed SR technique relies on GP to determine functional forms by interpolating through the given data points. SR does not possess any memory in the form of trained weights that can be applied onto other datasets with different distributions, and so, one needs to execute it separately in such a case. The solution process is more of an extraction rather than training.

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
