## [Peer Review File · Royal Society Open Science]

Review History

RSOS-211260.R0 (Original submission)

Review form: Reviewer 1

Is the manuscript scientifically sound in its present form?

No

Are the interpretations and conclusions justified by the results?

No

Is the language acceptable?

Yes

Do you have any ethical concerns with this paper?

No

Have you any concerns about statistical analyses in this paper?

No

Recommendation?

Reject

Comments to the Author(s)

Some minor grammatical errors throughout, not all given here, but two examples: p. 3 of 14, "blood flow through heart" should be "blood flow through the heart", "the novelty lied" should be "the novelty lay". Articles ("the", "a") are omitted in many places where they should be included.

p. 4 of 14 line 17/18 The paper talks about "the current FSI problem" without having explained what that problem is. Whether or not the problem is "computationally intractable" is subjective and highly dependent on the problem being considered.

Figure 1 Given the flow and deformation is 2D, the 3D nature of the box around the nodes could be slightly confusing. Is the domain periodic in all directions, or only the transverse (cross-flow) direction? If the latter, what boundary condition is used at the outlet? What flow speed is used, and what is the associated Reynolds number? Is the flow around the beam expected to be steady or unsteady?

p. 6 of 14 line 18/19 "The direction of fluid flow is orthogonal to the beam's longitudinal axis." This seems to contradict figure 1, where the beam is drawn at an oblique angle to the flow?

p. 7 of 14 Using structural dissipation to mimic the effects of fluid forces on the structure may work in some applications but would not be of general application. In particular the forces here depend only on the position and velocity of the structure, not the acceleration. What does this imply about the range of FSI problems that may be considered (e.g. steady/unsteady, large/small deflection)?

p. 7 of 14 line 55 and throughout the paper, details should be stated in non-dimensional fashion instead of dimensional. What is the characteristic time associated with the flow (e.g. frequency if vortex shedding off the beam as a bluff body), and how does the time step of 5×10^{-5} seconds compare to this? The same question for total run time t_f . Table 2 presents deformation data in m, which would be better presented non-dimensionalised by the beam length so that the physical significant can be better judged.

p. 9 of 14 line 11/12 What is f , and how does it relate to g ?

p. 10 of 14 line 20/21 Three beam dimensions are given, yet the model is only 2D?

Figures 3 and 4 Presentation of the data in this form gives a good overview, but makes detailed comparison difficult. The force predictions appear close in terms of the number and location of the various peaks, whereas the moment predictions do not (despite the authors' subjective claim that the fit is "descent"). Can the authors please comment on the reasons for accuracy in force and not in moment?

Figure 5 "referance" should be "reference".

Having read the paper, it is still not clear to me how the ML method can replace traditional FSI approaches (whether modelled with high fidelity, or via a reduced order model such as used here by approximation of the fluid forces via structural dissipation). As far as I can see the ML method has been used to generate a fitted surface representation of the solution as a function of x and t (Figures 3 and 4). What about prediction outside the given data set (either extending the time axis, or considering a case on which the ML algorithm was not trained)?

Review form: Reviewer 2

Is the manuscript scientifically sound in its present form?

No

Are the interpretations and conclusions justified by the results?

Yes

Is the language acceptable?

Yes

Do you have any ethical concerns with this paper?

No

Have you any concerns about statistical analyses in this paper?

No

Recommendation?

Major revision is needed (please make suggestions in comments)

Comments to the Author(s)

Dear Authors,

Please see my comments and questions in the attached document (Appendix A). Thank you.

Decision letter (RSOS-211260.R0)

Dear Dr Sarkar

The Editors assigned to your paper RSOS-211260 "Capturing functional relations in fluid-structure interaction via machine learning" have made a decision based on their reading of the paper and any comments received from reviewers.

Regrettably, in view of the reports received, the manuscript has been rejected in its current form. However, a new manuscript may be submitted which takes into consideration these comments.

We invite you to respond to the comments supplied below and prepare a resubmission of your manuscript. Below the referees' and Editors' comments (where applicable) we provide additional requirements. We provide guidance below to help you prepare your revision.

Please note that resubmitting your manuscript does not guarantee eventual acceptance, and we do not generally allow multiple rounds of revision and resubmission, so we urge you to make every effort to fully address all of the comments at this stage. If deemed necessary by the Editors, your manuscript will be sent back to one or more of the original reviewers for assessment. If the original reviewers are not available, we may invite new reviewers.

Please resubmit your revised manuscript and required files (see below) no later than 31-Mar-2022. Note: the ScholarOne system will 'lock' if resubmission is attempted on or after this deadline. If you do not think you will be able to meet this deadline, please contact the editorial office immediately.

Please note article processing charges apply to papers accepted for publication in Royal Society Open Science (<https://royalsocietypublishing.org/rsos/charges>). Charges will also apply to papers transferred to the journal from other Royal Society Publishing journals, as well as papers submitted as part of our collaboration with the Royal Society of Chemistry (<https://royalsocietypublishing.org/rsos/chemistry>). Fee waivers are available but must be requested when you submit your manuscript (<https://royalsocietypublishing.org/rsos/waivers>).

Thank you for submitting your manuscript to Royal Society Open Science and we look forward to receiving your resubmission. If you have any questions at all, please do not hesitate to get in touch.

on behalf of R. Kerry Rowe (Subject Editor)
openscience@royalsociety.org

Associate Editor Comments to Author:

Comments to the Author:

Two reviewers have offered views on your work. As you'll see, there is some divergence in opinion regarding your study - both in terms of its utility and the strength of the evidence you provide to support your conclusions. That said, it also seems to be the case that your work may have merit in a developing field. Given these viewpoints, we'd like you to prepare a substantially revised version of your paper that takes into account the commentary made - if you resubmit, the journal will invite these reviewers to reassess the paper. In the event that they are not satisfied by the changes made, it is likely your work will be rejected at that stage, so please do take care to fully address the reviewer feedback: clearly indicate changes made in a tracked-changes version of the revision and also in a point-by-point response.

Reviewer comments to Author:

Reviewer: 1

Comments to the Author(s)

Some minor grammatical errors throughout, not all given here, but two examples: p. 3 of 14, "blood flow through heart" should be "blood flow through the heart", "the novelty lied" should be "the novelty lay". Articles ("the", "a") are omitted in many places where they should be included.

p. 4 of 14 line 17/18 The paper talks about "the current FSI problem" without having explained what that problem is. Whether or not the problem is "computationally intractable" is subjective and highly dependent on the problem being considered.

Figure 1 Given the flow and deformation is 2D, the 3D nature of the box around the nodes could be slightly confusing. Is the domain periodic in all directions, or only the transverse (cross-flow) direction? If the latter, what boundary condition is used at the outlet? What flow speed is used,

and what is the associated Reynolds number? Is the flow around the beam expected to be steady or unsteady?

p. 6 of 14 line 18/19 "The direction of fluid flow is orthogonal to the beam's longitudinal axis." This seems to contradict figure 1, where the beam is drawn at an oblique angle to the flow?

p. 7 of 14 Using structural dissipation to mimic the effects of fluid forces on the structure may work in some applications but would not be of general application. In particular the forces here depend only on the position and velocity of the structure, not the acceleration. What does this imply about the range of FSI problems that may be considered (e.g. steady/unsteady, large/small deflection)?

p. 7 of 14 line 55 and throughout the paper, details should be stated in non-dimensional fashion instead of dimensional. What is the characteristic time associated with the flow (e.g. frequency if vortex shedding off the beam as a bluff body), and how does the time step of 5×10^{-5} seconds compare to this? The same question for total run time t_f . Table 2 presents deformation data in m, which would be better presented non-dimensionalised by the beam length so that the physical significant can be better judged.

p. 9 of 14 line 11/12 What is f , and how does it relate to g ?

p. 10 of 14 line 20/21 Three beam dimensions are given, yet the model is only 2D?

Figures 3 and 4 Presentation of the data in this form gives a good overview, but makes detailed comparison difficult. The force predictions appear close in terms of the number and location of the various peaks, whereas the moment predictions do not (despite the authors' subjective claim that the fit is "descent"). Can the authors please comment on the reasons for accuracy in force and not in moment?

Figure 5 "referance" should be "reference".

Having read the paper, it is still not clear to me how the ML method can replace traditional FSI approaches (whether modelled with high fidelity, or via a reduced order model such as used here by approximation of the fluid forces via structural dissipation). As far as I can see the ML method has been used to generate a fitted surface representation of the solution as a function of x and t (Figures 3 and 4). What about prediction outside the given data set (either extending the time axis, or considering a case on which the ML algorithm was not trained)?

Reviewer: 2

Comments to the Author(s)

Dear Authors,

Please see my comments and questions in the attached document. Thank you.

===PREPARING YOUR MANUSCRIPT===

===PREPARING YOUR REVISION IN SCHOLARONE===

- If you are requesting a discretionary waiver for the article processing charge, the waiver form must be included at this step.
- If you are providing image files for potential cover images, please upload these at this step, and inform the editorial office you have done so. You must hold the copyright to any image provided.
- A copy of your point-by-point response to referees and Editors. This will expedite the preparation of your proof.

- Ensure that your data access statement meets the requirements at <https://royalsociety.org/journals/authors/author-guidelines/#data>. You should ensure that you cite the dataset in your reference list. If you have deposited data etc in the Dryad repository, please include both the 'For publication' link and 'For review' link at this stage.
- If you are requesting an article processing charge waiver, you must select the relevant waiver option (if requesting a discretionary waiver, the form should have been uploaded at Step 3 'File upload' above).
- If you have uploaded ESM files, please ensure you follow the guidance at <https://royalsociety.org/journals/authors/author-guidelines/#supplementary-material> to include a suitable title and informative caption. An example of appropriate titling and captioning may be found at https://figshare.com/articles/Table_S2_from_Is_there_a_trade-off_between_peak_performance_and_performance_breadth_across_temperatures_for_aerobic_scoppe_in_teleost_fishes_/3843624.

Author's Response to Decision Letter for (RSOS-211260.R0)

See Appendix B.

RSOS-220097.R0

Review form: Reviewer 1

Is the manuscript scientifically sound in its present form?

Yes

Are the interpretations and conclusions justified by the results?

Yes

Is the language acceptable?

Yes

Do you have any ethical concerns with this paper?

No

Have you any concerns about statistical analyses in this paper?

No

Recommendation?

Accept as is

Comments to the Author(s)

The authors have addressed the concerns raised in my original review.

Review form: Reviewer 2

Is the manuscript scientifically sound in its present form?

Yes

Are the interpretations and conclusions justified by the results?

Yes

Is the language acceptable?

Yes

Do you have any ethical concerns with this paper?

No

Have you any concerns about statistical analyses in this paper?

No

Recommendation?

Accept with minor revision (please list in comments)

Comments to the Author(s)

Dear Authors,

I want to thank the authors for addressing all of my comments with thoughtful responses. In addition I found the revised figures and newly added appendices to be beneficial to the reader. I still have a few questions that I believe you could help clarify that I think would help better motivate some particular aspects of their study as well as strengthen it (see Appendix C).

Decision letter (RSOS-220097.R0)

Dear Dr Sarkar

On behalf of the Editors, we are pleased to inform you that your Manuscript RSOS-220097 "Capturing functional relations in fluid-structure interaction via machine learning" has been accepted for publication in Royal Society Open Science subject to minor revision in accordance

with the referees' reports. Please find the referees' comments along with any feedback from the Editors below my signature.

Please submit your revised manuscript and required files (see below) no later than 7 days from today's (ie 17-Feb-2022) date. Note: the ScholarOne system will 'lock' if submission of the revision is attempted 7 or more days after the deadline. If you do not think you will be able to meet this deadline please contact the editorial office immediately.

on behalf of Prof R. Kerry Rowe (Subject Editor)
openscience@royalsociety.org

Associate Editor Comments to Author:

Well done on largely resolving the queries/commentary from the referees - one has a few outstanding queries that we'd like you to address (see attached), but otherwise, good job.

Reviewer comments to Author:

Reviewer: 1

Comments to the Author(s)

The authors have addressed the concerns raised in my original review.

Reviewer: 2

Comments to the Author(s)

Dear Authors,

I want to thank the authors for addressing all of my comments with thoughtful responses. In addition I found the revised figures and newly added appendices to be beneficial to the reader. I still have a few questions that I believe you could help clarify that I think would help better motivate some particular aspects of their study as well as strengthen it.

===PREPARING YOUR MANUSCRIPT===

one version should clearly identify all the changes that have been made (for instance, in coloured highlight, in bold text, or tracked changes);

===PREPARING YOUR REVISION IN SCHOLARONE===

-- If you are requesting an article processing charge waiver, you must select the relevant waiver option (if requesting a discretionary waiver, the form should have been uploaded, see 'File upload' above).

-- If you have uploaded any electronic supplementary (ESM) files, please ensure you follow the guidance at <https://royalsociety.org/journals/authors/author-guidelines/#supplementary-material> to include a suitable title and informative caption. An example of appropriate titling and captioning may be found at https://figshare.com/articles/Table_S2_from_Is_there_a_trade-off_between_peak_performance_and_performance_breadth_across_temperatures_for_aerobic_sc_ope_in_teleost_fishes_/3843624.

Author's Response to Decision Letter for (RSOS-220097.R0)

See Appendix D.

Decision letter (RSOS-220097.R1)

Dear Dr Sarkar,

I am pleased to inform you that your manuscript entitled "Capturing functional relations in fluid-structure interaction via machine learning" is now accepted for publication in Royal Society Open Science.

on behalf of Professor R. Kerry Rowe (Subject Editor)
openscience@royalsociety.org

Appendix A

Capturing functional relations in fluid-structure interaction via machine learning

T. Soni, A. Sharma, R. Dutta, A. Dutta, S. Jayavelu, S. Sarkar

Overview

As the authors mention, FSI problems are ubiquitous in nature but are time consuming and difficult to perform. In this work, they developed a machine learning (ML) based strategy that harnesses symbolic regression (SR) that is able to capture the forces and moments of an immersed beam in background fluid flow. They were able to accurately capture the spatio-temporal dynamics, even across different discretizations of the beam. I believe this work is an important step forward in striving towards reducing the traditional overhead and time involved with using FSI analysis for engineering design and development. Novel ML-based approaches, like the one presented here, are valuable for helping guide the future of FSI analysis. I really enjoyed reading this manuscript, but I do have a few questions and (hopefully constructive) feedback for the authors.

Comments/Questions for the Authors:

- **(Section 2a)** Figure 1 was very helpful in gaining an idea of what is happening physically during the FSI simulations performed. However, I have a few clarifying questions:
 - Is the beam originally configured at an angle to the horizontal inflow? Figure 5 suggests that the beam is originally vertical oriented, where the incoming flow is perpendicular to it (since the deflections appear approximately symmetric about the middle of the beam)? If so, it may be worth mentioning and/or modifying Figure 1 to reflect the geometric configuration.
 - To clarify, is the beam tethered down at its end points? Or is it allowed to freely deform while getting pushed rightward to the left-to-right flow? The beam depiction in Figure 2 suggests the ends are tethered, possibly?
- **(Section 2a)** The authors mention that the fluid domain is of size $\Omega = 15\text{m} \times 15\text{m}$, but that the grid resolution is only 32×20 grid points. Unless, I am misinterpreting something (and I apologize if I am), that seems like a particularly low grid resolution for an FSI simulation. Could the authors confirm that this was the correct resolution used, or comment on their choice to use such a low resolution (was it sufficient, etc.)? If I misinterpreted this statement about the grid resolution used, it might be beneficial for readers to rephrase the statement. Thank you.

- Furthermore, I am confused as to why the square Eulerian (fluid) domain does not have equal resolution in both horizontal and vertical directions. Could the authors clarify this? Thanks.
- **(Section 5)** The authors mention that their framework is able capture the beam deflections for different FE discretizations. In what I've seen in the IB framework, the Eulerian and Lagrangian discretizations traditionally vary together, i.e., an Eulerian discretization (dx) is set and then an appropriate Lagrangian discretization (ds) is made (usually $ds \sim 0.5dx$ [Peskin, Acta Numerica, 2002]). For these higher FE discretizations of the beam, was the Eulerian (fluid) resolution also varied?
 - **(Sections 2a and 5)** To my above question, when varying the FE discretization for the IB2d simulation, how were the material property parameters of the beam changed (i.e., the beam stiffness, K_B , and axial (spring) stiffness, K_S)?
- **(Sections 2a)** Moreover, I realized that none of the simulation parameters are offered to the reader, even though the IB2d software used for the FSI simulations is open-source. A few other questions/comments then came to mind:
 1. Are the simulations performed here done with an existing example already available in the software? From glancing through the open-source examples, there appear to be many that involve background flows and/or bending/stretching beams, where the immersed structures are comprised of springs and beams, like the model here.
 2. Since the IB2d software is used for all of the FSI simulations performed here that serve as the data creation of the manuscript, it appears the authors only referenced one of the three manuscripts that the open-source IB2d software suggested to cite.
 3. Could the authors provide more detail on the properties of their beam as well as the inflow itself (e.g., was the inflow simply set to 0.01 m/s as mentioned in Section 4(a) or was it ramped up during the start of the simulation)? As ML-based techniques are being ever-more widely used, reproducibility is a cornerstone that must be considered [Brunton et al., Machine Learning for Fluid Mechanics, Annual Review of Fluid Mechanics, 2020 (<https://www.annualreviews.org/doi/full/10.1146/annurev-fluid-010719-060214>)]. Providing more such details will help bridge some gaps. I believe providing such details in an appendix would be appropriate, as the manuscript itself is already densely packed (but well-written and well-structured).
- **(Section 5)** The authors mention future works involve either other 2D problems or extending this framework to 3D. While I am excited to see such future works, I am wondering what some of the already known limitations might be, from having already perform the study here within the manuscript. Could the authors comment on any such limitations of using this ML-SR- based approach for uncovering FSI information that may have arose?
 - I know the authors had briefly commented in **Section 4(a)** that one challenge is fine tuning some of the parameters involved in the SR, so they use a grid fitting algorithm to uncover some.

However, I am curious about some of the other tuning parameters that were used and how sensitive the SR fit is them, e.g., the total mutation parameter, `p_mutation`: `p_subtree_mutation`, `p_hoist_mutation`, and `p_point_mutation`. Could the authors comment on how such values were chosen or how sensitive the fit is to these parameters?

Appendix B

Capturing functional relations in fluid-structure interaction via machine learning

Tejas Soni, Ashwani Sharma, Rajdeep Dutta, Annweshia Dutta, Senthilnath Jayavelu, Saikat Sarkar

(Dated: January 25, 2022)

We thank the editor and the reviewers for their helpful comments and insightful suggestions, according to which the manuscript has been amended. Kindly go through the following responses to the reviewer comments. The comments are shown in red color, their responses are shown in black color, and the changes in the manuscript are highlighted in blue color.

Reviewer 1 Comments and Responses

1. **Some minor grammatical errors throughout, not all given here, but two examples: p. 3 of 14, "blood flow through heart" should be "blood flow through the heart", "the novelty lied" should be "the novelty lay". Articles ("the", "a") are omitted in many places where they should be included.**

Response and Action: Thank you for pointing this out. We have thoroughly checked the grammar in the revised manuscript.

2. **p. 4 of 14 line 17/18 The paper talks about "the current FSI problem" without having explained what that problem is. Whether or not the problem is "computationally intractable" is subjective and highly dependent on the problem being considered.**

Response: We sincerely thank the reviewer for pointing this out. We realize the terminology, the current FSI problem, had been used before defining it, which actually appeared in the next section. The statement, 'computationally intractable' might also be misleading. However, we intended to mean that the computational requirement of the traditional FSI analysis can be drastically reduced by our proposed ML-based approach. Computational time taken by our ML-based approach : time taken by the traditional FSI analysis \approx 1:15.

Action: We have revised the manuscript to remove the confusions (please see the highlighted portions on page 3).

3. **Figure 1 Given the flow and deformation is 2D, the 3D nature of the box around the nodes could be slightly confusing. Is the domain periodic in all directions, or only the transverse (cross-flow) direction? If the latter, what boundary condition is used at the outlet? What flow speed is used, and what is the associated Reynolds number? Is the flow around the beam expected to be steady or unsteady?**

Response and Action: Thank you for pointing this out. We realize that the 3D box containing beam nodes may be confusing. We have removed this box from Figure 1 in the revised manuscript.

The fluid domain is periodic in all the direction, as considered in the work of Lai and Peskin, 2000. We have included this information in the revised manuscript. Their modelling approach relies on the fact that the closed form integral of the square of a spatio-temporal transport function is conserved for arbitrary fluid velocity under periodic boundary condition. (Please see the highlighted portion on page 4.)

In our study, water is used as the fluid medium with its properties: Dynamic Viscosity $\mu = 0.798 \times 10^{-3}$ (Ns/m^2)(30degC) and Density $\rho = 997$ (kg/m^3). The domain has a constant inlet velocity $v = 1$ cm/sec with an associated Reynolds Number $Re = \frac{\rho v d}{\mu} = 187406$. Note that the Reynolds number changes (locally) around the beam due to the fluid velocity variations. The fluid velocity around the beam varies from 4×10^{-6} to 0.045 cm/sec, indicating a range of Re : 75 - 850823. This reveals that the nature of the flow around the beam varies from steady to unsteady. (Please see the highlighted portion on page 4 of the revised manuscript.)

- (a) Lai Ming-Chih, Peskin CS . 2000 An immersed boundary method with formal second-order accuracy and reduced numerical viscosity. *Journal of computational Physics*, **160.2**, 705-719.

4. p. 6 of 14 line 18/19 "The direction of fluid flow is orthogonal to the beam's longitudinal axis." This seems to contradict figure 1, where the beam is drawn at an oblique angle to the flow?

Response and Action: We thank the reviewer for pointing this out. In the revised manuscript, to maintain consistency, we have amended Figure 1 depicting an immersed beam placed orthogonally with reference to the direction of fluid flow. The amended figure is shown below.

Figure 1: A simply supported beam immersed inside a fluid domain with constant flow at the inlet: the beam is placed orthogonally with reference to the direction of the fluid flow; the involved coordinate systems are shown in a generic blue and the nodal deflections are shown by highlighting three nodes of the beam.

5. p. 7 of 14 Using structural dissipation to mimic the effects of fluid forces on the structure may work in some applications but would not be of general application. In particular the forces here depend only on the position and velocity of the structure, not the acceleration. What does this imply about the range of FSI problems that may be considered (e.g. steady/unsteady, large/small deflection)?

Response: Thank you for raising this point. To capture the FSI effect, in the current work, we have considered Langevin equation without including the inertial effect. Note that the Langevin equation (without the inertia term) is typically used to understand the mechanics of bio-molecules and many such systems immersed in fluid. It is interesting to see that we are able to capture the structural vibration reasonably well using the dissipative beam model, while neglecting the inertia term. However, we would like to extend the present work elsewhere by incorporating the inertial effect as well in the dissipative reduced dimensional model to better capture (particularly for the higher modes) the deformation of the immersed structure immersed.

6. p. 7 of 14 line 55 and throughout the paper, details should be stated in non-dimensional fashion instead of dimensional. What is the characteristic time associated with the flow (e.g. frequency if vortex shedding off the beam as a bluff body), and how does the time step of 5×10^{-5} seconds compare to this? The same question for total run time t_f . Table 2 presents deformation data in m, which would be better presented non-dimensionalised by the beam length so that the physical significance can be better judged.

Response and Action: Thanks for your suggestion. In this work, we have considered a simply supported beam with the modulus of elasticity $E = 2 \times 10^{11} \frac{N}{m^2}$, area moment of inertia $I = 2.133 \times 10^{-10} m^4$ and linear mass density, $\rho_L = 8050 \times 0.008 \times 0.005 \frac{Kg}{m}$, which refers to the (first) natural frequency of vibration as: $f_0 = \frac{1}{2\pi} \left(\frac{\pi}{L}\right)^2 \sqrt{\frac{EI}{\rho_L}} = 0.1808$ Hz and the corresponding time period is: $T_0 = \frac{1}{f} = 5.53$ sec. Since the present

dissipative beam deformation mimics an overdamped system dynamics, the damped frequency is < 0.1808 Hz and the corresponding time period > 5.53 sec. The characteristic time is chosen as $T_c = 15$ sec as we want to observe the vibration response for a long time span. The ratio of $dt : T_C = 1 : 300000$, where t_f is same as T_c .

Earlier research works provided different non-dimensionalization schemes suitable to their applications [(a) Bhattacharya and Adhikari, (b) Akella et al.]. For instance, Akella et al. adopted a procedure to non-dimensionalize the governing E-B equation of motion by: $\tilde{x} = \frac{x}{L}$ and $\tilde{t} = \frac{t}{L^2} \sqrt{\frac{EI}{\rho L}} = \frac{t}{(\pi/2)*T_0}$, while considering free vibration.

In our research, we have non-dimensionalized the associated independent axes as: $\tilde{x} = \frac{x}{L}$, $\tilde{t} = \frac{t}{T_c}$. In the revised manuscript, we have used non-dimensional independent axes in Figures 3, 4, and 5. Also, Table 2 has been updated to reflect the deformation differences as fraction (percentage) of the beam length.

- (a) Adhikari, Sondipon, Bhattacharya, Subhamoy. 2011 Vibrations of wind-turbines considering soil-structure interaction, *Wind and Structures*, 85.
- (b) Akella, Prithvi and Hemingway, Evan G and O'Reilly, Oliver M. A Visualization Tool for the Vibration of Euler-Bernoulli and Timoshenko Beams.

7. **p. 9 of 14 line 11/12 What is f , and how does it relate to g ?**

Response: Thank you for mentioning this. We realize that it's a mistake and $f(\chi)$ must be replaced with $g(\chi)$.

Action: The revised manuscript uses $g(\chi)$ consistently in Section 3(b) to represent a generic mapping/function approximated by the symbolic regression (SR) technique.

8. **p. 10 of 14 line 20/21 Three beam dimensions are given, yet the model is only 2D?**

Response: Thanks for stating the confusion. In our work, three beam dimensions (length 10m, width 5mm and depth 8mm). The beam's width and depth are negligible as compared to its length with an aspect ratio $\gg 10$. Thus the structure falls under the realm of Euler-Bernoulli (E-B) beam. The beam's width and depth are used to calculate the moment of inertia, I (in Equation 2.3 of the revised manuscript). This E-B beam is submersed into a two-dimensional fluid flow, where its length is orthogonal to the direction of the fluid flow.

9. **Figures 3 and 4 Presentation of the data in this form gives a good overview, but makes detailed comparison difficult. The force predictions appear close in terms of the number and location of the various peaks, whereas the moment predictions do not (despite the authors' subjective claim that the fit is "descent"). Can the authors please comment on the reasons for accuracy in force and not in moment?**

Response: Thank you for raising this question on the inference from results.

Figure A: Beam deflection and rotation profiles at three time instants.

Action: Regarding this, the following discussion has been included in the revised manuscript on page 11 (just before Section 4(b)).

The moment is applied onto the rotational degrees of freedom, whereas the force is applied onto the translational degrees of freedom. As the rotations are more undulating than the displacements, intuitively, we expect the

moment landscape to be more irregular than the force landscape with several peaks and valleys. Therefore, the currently employed ML technique, i.e. symbolic regression (SR), confronts challenges to capture the spatio-temporal functional relations in the moment profile. To tackle this high complexity, we tried to empower SR by feeding it more basis (elementary composition) functions. Still, the accuracy of the achieved moment fit is slightly worse than that of the force fit; however, the reported result is the best among various solutions obtained with different parameter settings in SR.

Figure A justifies the fact that the rotations are more undulating than the displacements.

10. **Figure 5 "referance" should be "reference".**

Response and Action: We are extremely sorry for this typo. The same has been corrected in Figure 5 of the revised manuscript.

11. **Having read the paper, it is still not clear to me how the ML method can replace traditional FSI approaches (whether modelled with high fidelity, or via a reduced order model such as used here by approximation of the fluid forces via structural dissipation).**

Response: In conventional FSI analysis, the interaction principle relies on the force transfer from structure to fluid and the velocity transfer from fluid to structure, which involves computationally expensive numerical integrations for calculating structural displacements. However, in our approach, the fluid and structural domains are segregated and structural displacements are calculated by feeding the ML-generated forces and moments into the dissipative (damped) structural model. Thus, this work proposes the immersed beam's displacement calculations, while alleviating the computational overhead involved in solving Navier-Stokes by mimicking the fluid flow effects with ML-generated forcing functions into a dissipative EB beam model.

Action: The revised manuscript includes the above information in a qualitative comparison table (Table 1) on page 8.

Table 1: A qualitative comparison between the existing and our proposed FSI analysis approaches to highlight some of the key factors and advantages.

Attributes	Traditional FSI Analysis	Our ML-based Approach
Navier-Stokes Solution Handling	Explicitly solves Navier-Stokes equations governing fluid motions, which involves huge computational overhead.	Bypasses solving Navier-Stokes by mimicking fluid flow effects with proper forcing functions into dissipative Euler-Bernoulli beam model.
Structural Displacement Calculation	Displacements are calculated via numerical integration involving force and velocity transfers from structure and fluid and vice versa.	Displacements are calculated by feeding SR-generated forces and moments into the dissipative Euler-Bernoulli beam model.
Cost of Computation	Computationally expensive due to simultaneously solving numerical integration .	Computationally inexpensive due to elegant functional forms of forces/moments.

As far as I can see the ML method has been used to generate a fitted surface representation of the solution as a function of x and t (Figures 3 and 4). What about prediction outside the given data set (either extending the time axis, or considering a case on which the ML algorithm was not trained)?

Response and Action: The revised manuscript includes an Appendix, where we have mentioned the following information under the subsection Extrapolation Capability.

The employed ML technique, symbolic regression (SR), fits continuous symbolic functions passing through the given data points. Thus, the achieved outcome expressions have adequate interpolation capability, as supported by the performance evaluation study where the same functional forms of forces/moments work under different FE discretizations in Table 2. We are able to extrapolate the ML-solutions to some extent, however, the extrapolation ability of these expressions still remains a concern. The current focus is on capturing the spatio-temporal patterns within a small scale. In future, we would like to extend our approach to a large scale while neglecting high resolution spatio-temporal oscillations.

It is challenging for the ML-based solver to fit extrapolated data due to two factors: (i) the innate limitation of the fitting approach by SR, (ii) the transient behavior of the vibration response. In case the length of the beam is increased while other parameters remain constant, then the extrapolation can work within a small range, as justified by Figures 7a and 7b. In Figure 7a, the distribution (spatio-temporal patterns) of force do not change much for beam lengths 9, 10, 11, 12 m , and in Figure 7b, the distribution of moment do not change much for beam lengths 9, 10 m and for 11, 12 m ; only their magnitudes change to some extent. So, the same force and moment functional forms generated by SR for 10 m beam can extrapolate to 9 – 12 m beams in combination

with proper scaling factors. Further, the spatio-temporal patterns of the force and moment landscapes remain similar when the fluid velocity (at the inlet) is varied within a range of 0.005 – 0.04 m/sec for a 10 m beam, as depicted in Figures 8a and 8b. The force and moment patterns are very close for a velocity range of 0.005 – 0.04 m/sec , except a few peaks in the forces start growing when velocity is more than 0.02 m/sec . Figure 9 shows that the force and moment expressions obtained for a 10 m beam, along with the respective scaling factors (1.07, 0.91), can reconstruct the deflections of a 11 m beam for 5 secs. Thus, the extrapolation is possible for a range of velocities as well. Figure 9 also shows that the force and moment expressions obtained for a 10 m beam immersed into a fluid flow of inlet velocity 0.01 m/sec , along with the respective scaling factors (2, 2), can reconstruct the deflections under different inlet velocity 0.02 m/sec for 5 secs. Nevertheless, the extrapolation ability along the time axis is challenging, which we plan to tackle in future by feeding data that spans over a long time.

Unlike neural network (NN) based ML techniques, the currently employed SR technique relies on genetic programming to determine functional forms by interpolating through the given data points. SR does not possess any memory in the form of trained weights that can be applied onto other data sets with different distributions, and so, one needs to execute it separately in such a case. The solution process is more of an extraction rather than training.

Reviewer 2 Comments and Responses:

Overview: As the authors mention, FSI problems are ubiquitous in nature but are time consuming and difficult to perform. In this work, they developed a machine learning (ML) based strategy that harnesses symbolic regression (SR) that is able to capture the forces and moments of an immersed beam in background fluid flow. They were able to accurately capture the spatio-temporal dynamics, even across different discretizations of the beam. I believe this work is an important step forward in striving towards reducing the traditional overhead and time involved with using FSI analysis for engineering design and development. Novel ML-based approaches, like the one presented here, are valuable for helping guide the future of FSI analysis. I really enjoyed reading this manuscript, but I do have a few questions and (hopefully constructive) feedback for the authors.

Comments/Questions for the Authors:

1. (Section 2a) Figure 1 was very helpful in gaining an idea of what is happening physically during the FSI simulations performed. However, I have a few clarifying questions: – Is the beam originally configured at an angle to the horizontal inflow?

Response and Action: We thank the reviewer for the comments. The beam is originally orthogonal to the fluid flow. We have amended Figure 1 accordingly to avoid confusion.

Note: We provide a supplementary file where the ground truth data (force and moment) is generated for a beam immersed into the water at an inclination to the direction of the flow.

2. Figure 5 suggests that the beam is originally vertical oriented, where the incoming flow is perpendicular to it (since the deflections appear approximately symmetric about the middle of the beam)? If so, it may be worth mentioning and/or modifying Figure 1 to reflect the geometric configuration.

Response and Action: We thank the reviewer for pointing this out. In the revised manuscript, to maintain consistency with our simulation, we have amended Figure 1 depicting an immersed beam placed orthogonally with reference to the direction of the fluid flow and indicated its geometric configuration (orthogonal) in the figure caption. The amended figure is shown below.

3. To clarify, is the beam tethered down at its end points? Or is it allowed to freely deform while getting pushed rightward to the left-to-right flow? The beam depiction in Figure 2 suggests the ends are tethered, possibly?

Response: Thank you for pointing this out. Yes, the end points of the beam are tethered, i.e. simply supported. The end point conditions of the beam were mentioned only in the caption of Figure 1, which was inadequate to clarify the underlying boundary conditions.

Action: In Section 2(b) on page 5 of the revised manuscript, we include a brief description on the end-point condition of the beam. The immersed beam shown in Figure 1 is simply supported (hinged) at both the ends, which restricts the translational motion but allows the rotational motion of the end points. According to the associated boundary conditions, i.e. $w(0, t) = w(L, t) = 0$ and $\frac{\partial^2 w(0, t)}{\partial x^2} = \frac{\partial^2 w(L, t)}{\partial x^2} = 0$, the end nodes of the

beam are stationary though the intermediate nodes get displaced due to the fluid flow, which in turn deforms the beam without pushing (moving) it along the flow direction.

Figure 1: A simply supported beam immersed inside a fluid domain with constant flow at the inlet: the beam is placed orthogonally with reference to the direction of the fluid flow; the involved coordinate systems are shown in a generic way and the nodal deflections are shown by highlighting three nodes of the beam.

4. (Section 2a) The authors mention that the fluid domain is of size $W = 15m \times 15m$, but that the grid resolution is only 32 X 20 grid points. Unless, I am misinterpreting something (and I apologize if I am), that seems like a particularly low grid resolution for an FSI simulation. Could the authors confirm that this was the correct resolution used, or comment on their choice to use such a low resolution (was it sufficient, etc.)? If I misinterpreted this statement about the grid resolution used, it might be beneficial for readers to rephrase the statement. Thank you.

Response: Thanks for your concern. The answer to this question is given along with the answer to the next one (question 5).

5. Furthermore, I am confused as to why the square Eulerian (fluid) domain does not have equal resolution in both horizontal and vertical directions. Could the authors clarify this? Thanks.

Response and Action: Thanks for mentioning this. We have performed the simulation with different grid resolution, however, the one mentioned in the manuscript considered to reduce the computational overhead without compromising the accuracy. In support of our argument, a quantitative evaluation is added in Table A.

Table A: A performance evaluation comparison of different fluid domain grid resolutions, carried out with reference to the deflection simulated (time consumption= 70 mins) using very resolution grids: 100 grids along x-axis \times 100 grids along y-axis. The error metric (RMSE) is calculated between all the deflection points (65×3001) for 15 secs simulation while considering 65 beam nodes and a time increment of 5×10^{-3} sec.

Grid resolution	Time consumption (min)	Simulation accuracy (RMSE)
32 grids along x-axis \times 20 grids along y-axis	14	0.0189
20 grids along x-axis \times 32 grids along y-axis	14	0.0293
32 grids along x-axis \times 32 grids along y-axis	17	0.0310

6. (Section 5) The authors mention that their framework is able to capture the beam deflections for different FE discretizations. In what I've seen in the IB framework, the Eulerian and Lagrangian discretizations

traditionally vary together, i.e., an Eulerian discretization (dx) is set and then an appropriate Lagrangian discretization (ds) is made (usually $ds = 0.5dx$ [Peskin, *Acta Numerica*, 2002]). For these higher FE discretizations of the beam, was the Eulerian (fluid) resolution also varied?

Response and Action: We thank the reviewer for pointing this point. In this regard, we have included the following information in the revised manuscript (see page 12, Section 4b).

In this context, note that for the detailed FSI analysis, Lagrangian discretization (ds) is taken: $0.5 \times$ Eulerian discretization (dx), as considered in IB2d [11].

7. – (Sections 2a and 5) To my above question, when varying the FE discretization for the IB2d simulation, how were the material property parameters of the beam changed (i.e., the beam stiffness, KB , and axial (spring) stiffness, KS)?

Response and Action: The bending stiffness and axial stiffness of the beam increase as the number of discrete nodes increases. The axial stiffness increases at a linear rate, and the bending (flexural) stiffness increases at a nonlinear rate. Figure B shows different flexural stiffness profiles under various FE discretizations.

Figure B: Flexural stiffness profiles for different FE discretizations: $N = 65, 100, 200$.

8. (Sections 2a) Moreover, I realized that none of the simulation parameters are offered to the reader, even though the IB2d software used for the FSI simulations is open-source. A few other questions/comments then came to mind:

- (a) Are the simulations performed here done with an existing example already available in the software? From glancing through the open-source examples, there appear to be many that involve background flows and/or bending/stretching beams, where the immersed structures are comprised of springs and beams, like the model here.

Response and Action: We thank the reviewer for pointing this out. In the IB2d open source software, the spring stiffness is constant. However, we have considered variable stiffness values along the length of the beam (both axial and bending), calculated using the finite element method. This part does not exist in the IB2d open source code. In the revised version, we have mentioned all the necessary details for the FSI analysis.

- (b) Since the IB2d software is used for all of the FSI simulations performed here that serve as the data creation of the manuscript, it appears the authors only referenced one of the three manuscripts that the open-source IB2d software suggested to cite.

Response and Action: We thank the reviewer for bringing this to our notice. We apologize for the missing citations. In the revised manuscript, we have cited all the three IB2d related papers: (i) ‘*IB2d: a Python and MATLAB implementation of the immersed boundary method*’, (ii) ‘*IB2d Reloaded: A more powerful Python and MATLAB implementation of the immersed boundary method*’, and (iii) ‘*A mathematical model and matlab code for muscle-fluid-structure simulations*’.

- (c) Could the authors provide more detail on the properties of their beam as well as the inflow itself (e.g., was the inflow simply set to 0.01 m/s as mentioned in Section 4(a) or was it ramped up during the start of the simulation)?

Response and Action: Thanks for pointing this out. In the revised manuscript, we have included the properties of the beam and the fluid flow in Section 2(b) on page 5 and in in Section 2(a) on page 4, respectively.

Beam properties: In the present work, we consider a simply supported beam of length $L = 10\text{ m}$ with the properties: modulus of elasticity $E = 2 \times 10^{11}\text{ Nm}^{-2}$, area moment of inertia $I = 2.133 \times 10^{-10}\text{ m}^4$, and linear mass density $\rho_L = 8050 \times 0.008 \times 0.005\text{ Kgm}^{-1}$. The immersed beam shown in Figure 1 is simply supported (hinged) at both the ends, which restricts the translational motion but allows the rotational motion of the end points. According to the associated boundary conditions, i.e. $w(0, t) = w(L, t) = 0$ and $\frac{\partial^2 w(0, t)}{\partial x^2} = \frac{\partial^2 w(L, t)}{\partial x^2} = 0$, the end nodes of the beam are stationary though the intermediate nodes get displaced due to the fluid flow, which in turn deforms the beam without pushing (moving) it along the flow direction.

Fluid flow properties: The fluid domain is periodic in all the direction. In our study, water is used as the fluid medium with its properties: Dynamic Viscosity $\mu = 0.798 \times 10^{-3}\text{ (Ns/m}^2\text{)}(30\text{degC})$ and Density $\rho = 997\text{ (kg/m}^3\text{)}$. The domain has a constant inlet velocity $v = 0.01\text{ m/sec}$ with an associated Reynolds Number $Re = (\frac{\rho v d}{\mu}) = 187406$. Note that the Reynolds number changes (locally) around the beam due to the fluid velocity variations. The fluid inflow was kept constant and not ramped up during the start of the simulation.

As ML-based techniques are being ever-more widely used, reproducibility is a cornerstone that must be considered [Brunton et al., Machine Learning for Fluid Mechanics, Annual Review of Fluid Mechanics, 2020 (<https://www.annualreviews.org/doi/full/10.1146/annurevfluid-010719-060214>)]. Providing more such details will help bridge some gaps. I believe providing such details in an appendix would be appropriate, as the manuscript itself is already densely packed (but well-written and well-structured).

Response and Action: Thank you for commenting on the reproducibility of the ML-generated results. In the currently employed SR solver, i.e. gplearn, random numbers are used at two places. The first random number is used to apply a permutation for shuffling the input data points to the solver. The second random number is used in the genetic programming for candidate evolution. The SR-generated forces/moments are reproducible given that the random numbers are called from a fixed seed during its *python* implementation. This information is also added in the Appendix of the revised manuscript.

9. **(Section 5) The authors mention future works involving either other 2D problems or extending this framework to 3D. While I am excited to see such future works, I am wondering what some of the already known limitations might be, from having already performed the study here within the manuscript. Could the authors comment on any such limitations of using this ML-SR- based approach for uncovering FSI information that may have arisen?**

Response and Action: The revised manuscript includes an Appendix, where we have mentioned all the following points.

The currently employed ML technique, i.e. symbolic regression (SR), succeeds in capturing the spatio-temporal functional relations present in the ground truth forces/moments profiles. However, during the implementation, we confronted challenges as mentioned below.

- SR relies on genetic programming, which is very sensitive to the input parameters like ‘population size’ and ‘tournament size’. Hence, we had to perform a grid search for a proper parameter tuning.
- The choice of basis (elementary composition) functions is critical and important. In practise, we conducted many trial and error simulation runs before coming up with a suitable library of basis functions. The ground truth moment profile had more complexity than the force profile. To tackle this, we empowered SR by feeding it a slightly bigger library of basis functions.
- To avoid numerical overflow errors [39], we protected some of the basis functions like ‘sqrt’, ‘exp’ in genetic programming. This minute step is important in simulations.

In future, our proposed framework can be extended from 2D to 3D problems while carefully addressing two major factors, i.e. parameter sensitivity and the choice of basis functions, during the implementations.

The limitations of the proposed approach are discussed below, which are also included in Appendix under the subsection Extrapolation Capability.

- The employed ML technique, i.e. symbolic regression (SR), fits continuous symbolic functions that pass through the given data points. Thus, the achieved outcome expressions have adequate interpolation capability, as supported by the performance evaluation study under different FE discretizations in Table 2. We are able to extrapolate the ML-solutions to some extent, however, the extrapolation ability of these expressions still remains a concern. The current focus is on capturing the spatio-temporal patterns within a small scale. In future, we would like to extend our approach to a large scale while neglecting high resolution spatio-temporal oscillations.
- Unlike neural network (NN) based ML techniques, the currently employed SR technique relies on genetic programming to determine functional forms by interpolating through the given data points. SR does not possess any memory in the form of trained weights that can be applied onto other data sets with different distributions, and so, one needs to execute it separately in such a case. The solution process is more of an extraction rather than training.

10. – I know the authors had briefly commented in Section 4(a) that one challenge is fine tuning some of the parameters involved in the SR, so they use a grid fitting algorithm to uncover some. However, I am curious about some of the other tuning parameters that were used and how sensitive the SR fit is them, e.g., the total mutation parameter, p mutation: p subtree mutation, p hoist mutation, and p point mutation. Could the authors comment on how such values were chosen or how sensitive the fit is to these parameters?

Response and Action: Thank you for your suggestion. The following information has been added under 'Parameter Tuning' subheading in the Appendix of the revised manuscript.

Parameter tuning is an important aspect in Symbolic Regression (SR). SR relies on Genetic Programming, where a balance between exploration and exploitation needs to be maintained for optimal performance. To understand the choice of tuning parameters, first we need to realize their roles and purposes. Here, p.crossover parameter takes care of the exploitation and p_mutation=(p_subtree_mutation+p_hoist_mutation+p_point_mutation) takes care of the exploration of the search space. 'p_subtree_mutation' is an aggressive mutation parameter since it can replace significant genetic material, i.e. a subtree composed of nodes and branches, from the tournament winners to introduce extinct functions/operators into a population of candidate solutions; 'p_hoist_mutation' is a bloat-resisting mutation parameter that removes genetic material by hoisting a random subtree into the original subtree's location; and 'p_point_mutation' is a nominal mutation parameter that replaces random nodes from the tournament winners to introduce new function/operator [gplearn]. Figure 6 gives a visualization of the crossover and mutation operations.

Figure 6: A visualization of the crossover and mutation operations involved in the evolution of candidate solution trees.

Appendix C

Capturing functional relations in fluid-structure interaction via machine learning

T. Soni, A. Sharma, R. Dutta, A. Dutta, S. Jayavelu, S. Sarkar

Overview

I want to thank the authors for addressing all of my comments with thoughtful responses. In addition I found the revised figures and newly added appendices to be beneficial to the reader. I still have a few questions that I believe the authors can help clarify that I think would help better motivate some particular aspects of their study.

Comments/Questions for the Authors:

- (Table A (in author's response))
 - I suggest including this Table in the manuscript as it may be useful for readers with a computational fluid dynamics background. Readers with a CFD background they will likely wonder why these simulations were run with such low grid resolution. In addition, they will also likely wonder why the grid resolution in the x and y directions are not equal, i.e., $dx = L_x/N_x \neq dy = L_y/N_y$. Can the authors provide any argument for what the benefit of a non-uniform spatial resolution would be? Also, how would that affect the units of the parameters of the beam?
 - Just a small grammatical typo in the Table caption: please consider changing *very resolution grids* \Rightarrow *higher resolved grids*.
 - Could the authors provide a bit more detail in how they calculated the accuracy via the RSME. Are they comparing time-points to the same time-points and then averaging over the entire simulation? Also, what specifically are the authors comparing - the position of the beam's Lagrangian points? the force at each Lagrangian point? The overall force over the beam?
 - Also, was the number of Lagrangian points in the beam held constant among all cases (including the 100×100 case? If so, how does that satisfy the $ds = 0.5dx$ that the authors state their using? Or, if not, how are they scaling the beam's spring stiffnesses and beam stiffnesses to compensate for the change in grid discretization? This is also my question for Figure B that was provided.
- (Reynolds number details in manuscript) I appreciate the authors trying to provide a Reynolds number for this deformation system; however, I would avoid computing velocities near (or directly at) the location of the beam (immersed boundary) due to spurious oscillations that are known to occur in IB frameworks. If they wish to provide a Reynolds number, I might suggest to use the inflow velocity

and the size (length) of the beam itself as the characteristic velocity and characteristic length scale, respectively.

- **(Symbolic Regression)** I appreciate the authors provide details about SR in a newly added appendix; however, I have a few suggestions and comments for the authors that I believe would help strengthen their manuscript and place it in the context of other ML-based techniques used in FSI.
 - **(Introduction)** The authors introduce the notions of what other ML methods have been applied to FSI problems, like various flavors of neural networks (PINNs, RNNs, MLPs. etc.) and mention that their is still a high computational cost to generate the training data necessary for such ML approaches. However, could the authors provide the clear advantages of what SR offers over those other approaches? The authors mention that SR leverages one simulation to perform a fitting routine through the output data; however, extrapolation to other cases is not clear. Could the authors provide insight into what benefit SR has in these contexts? Or perhaps, what the end goal of SR would be, particularly if it is limited in extrapolating to other cases.
 - **(Table 1)** This table really helps provide context in regard to comparing the traditional approach to FSI and the authors' ML-based approach. With the reduced model given by SR, I struggle to be able to extract and physical meaning/intuition from Eqns. 4.1 and 4.2. In *Section 3B: ML for extracting functional forms*, the authors mention that SR offers *interpretability of the underlying physics*. Could the authors help clarify this or mention specifically how Eqns. 4.1 and 4.2 can be interpreted? Thank you!
- **(Table 3)** For the varying number of beam Lagrangian points in Table 3, am I correct in believe that the grid resolution is also changing? Can the authors clarify this, either in the Table's caption or in the text, as well as provide what grid resolutions those numbers of beam nodes correspond to. Moreover, how are the beam's stiffness parameters being scaled in this study? Also, could the authors clarify what the *reference* case is? Is this the higher resolution case mentioned earlier of 100×100 ?

Appendix D

Capturing functional relations in fluid-structure interaction via machine learning

Tejas Soni, Ashwani Sharma, Rajdeep Dutta, Annwasha Dutta, Senthilnath Jayavelu, Saikat Sarkar

(Dated: February 24, 2022)

We thank the editor and the reviewers for their helpful comments and insightful suggestions, according to which the manuscript has been amended. Kindly go through the following responses to the reviewer comments. The comments are shown in red color, their responses are shown in black color, and the changes in the manuscript are highlighted in blue color.

Reviewer Comments and Responses

- (a) **I suggest including this Table in the manuscript as it may be useful for readers with a computational fluid dynamics background. Readers with a CFD background they will likely wonder why these simulations were run with such low grid resolution. In addition, they will also likely wonder why the grid resolution in the x and y directions are not equal, i.e., $dx = L_x = N_x \neq dy = L_y = N_y$. Can the authors provide any argument for what the benefit of a non-uniform spatial resolution would be? Also, how would that affect the units of the parameters of the beam?**
- (b) **Just a small grammatical typo in the Table caption: please consider changing very resolution grids)higher resolved grids.**
- (c) **Could the authors provide a bit more detail in how they calculated the accuracy via the RSME. Are they comparing time-points to the same time-points and then averaging over the entire simulation? Also, what specifically are the authors comparing - the position of the beam's Lagrangian points? the force at each Lagrangian point? The overall force over the beam?**

Response and Action: Thank you for your valuable suggestions, according to which, we have further amended the manuscript. The above three comments are addressed and clarified in a new subsection *Fluid Domain Resolution* of the Appendix. For better readability, we also add the same answer in the following.

Fluid Domain Resolution:

We have performed the simulation with different grid resolutions, however, the resolution (32×20) has been considered to reduce the computational overhead without compromising the accuracy in deflections. In support of this argument, a quantitative evaluation study is presented in Table 4.

Table 4: A performance evaluation comparison of different fluid domain grid resolutions, carried out with reference to the deflection simulated using higher resolved grids: 100 grids along x-axis \times 100 grids along y-axis. The root mean square error (RMSE) is calculated between the deflections ($\in \mathbb{R}^{65 \times 3000}$) obtained with different grid resolutions, at the corresponding Lagrangian points (65) for the same time instances (3000).

Grid resolution	Time consumption (min)	Simulation error (RMSE)
100 grids along x-axis \times 100 grids along y-axis	70	0.00 (ref)
32 grids along x-axis \times 20 grids along y-axis	14	0.0189
20 grids along x-axis \times 32 grids along y-axis	14	0.0293
32 grids along x-axis \times 32 grids along y-axis	17	0.0310

- (d) **Also, was the number of Lagrangian points in the beam held constant among all cases (including the 100×100 case? If so, how does that satisfy the $ds = 0.5dx$ that the authors state their using? Or, if not, how are they scaling the beam's spring stiffnesses and beam stiffnesses to compensate for the change in grid discretization? This is also my question for Figure B that was provided.**

Response and Action: We sincerely thank the reviewer for pointing this out.

For the convergence studies, we have refined the Eulerian mesh while keeping the Lagrangian mesh invariant, which implies that the bending stiffness and the axial stiffness remain constant under different fluid domain resolutions. However, in our final simulation, the underlying idea was to use the Lagrangian mesh (beam) finer than the Eulerian mesh (fluid) for accurate visualization of the deformation profile of the beam.

2. (Reynolds number details in manuscript) I appreciate the authors trying to provide a Reynolds number for this deformation system; however, I would avoid computing velocities near (or directly at) the location of the beam (immersed boundary) due to spurious oscillations that are known to occur in IB frameworks. If they wish to provide a Reynolds number, I might suggest to use the inflow velocity and the size (length) of the beam itself as the characteristic velocity and characteristic length scale, respectively

Response and Action: Thanks again for your suggestion. Please see the highlighted portion on page 4 of the revised manuscript, as stated below.

In our study, water is used as the fluid medium with its properties: Dynamic Viscosity $\mu = 0.798 \times 10^{-3}$ (Ns/m^2)(30degC) and Density $\rho = 997$ (kg/m^3). The domain has a constant inlet velocity $v = 1$ cm/sec with an associated Reynolds Number $Re = (\frac{v d}{\mu}) = 187406$. Note that the Reynolds number changes (locally) around the beam due to the fluid velocity variations.

3. (Symbolic Regression) I appreciate the authors provide details about SR in a newly added appendix; however, I have a few suggestions and comments for the authors that I believe would help strengthen their manuscript and place it in the context of other ML-based techniques used in FSI.

- (a) (Introduction) The authors introduce the notions of what other ML methods have been applied to FSI problems, like various flavors of neural networks (PINNs, RNNs, MLPs. etc.) and mention that their is still a high computational cost to generate the training data necessary for such ML approaches. However, could the authors provide the clear advantages of what SR offers over those other approaches? The authors mention that SR leverages one simulation to perform a fitting routine through the output data; however, extrapolation to other cases is not clear. Could the authors provide insight into what benefit SR has in these contexts? Or perhaps, what the end goal of SR would be, particularly if it is limited in extrapolating to other cases.

Response and Action: Thank you for pointing out this. To motivate the use of SR over NN in the underlying FSI problem, we include the following sentence in the last paragraph of the Introduction.

An NN-based approximation relies on the combination of numerical weights and activations, whereas an SR-based approximation provides physically intuitive expression(s) by means of symbolic functions and operators [25].

The extrapolation capability of the SR has been already discussed the Appendix. Although the achieved solutions can be extrapolated to some extent (outside the regime of the training velocity and length), they unfold symbolic functions and operators that are physically meaningful.

- (b) (Table 1) This table really helps provide context in regard to comparing the traditional approach to FSI and the authors' ML-based approach. With the reduced model given by SR, I struggle to be able to extract and physical meaning/intuition from Eqns. 4.1 and 4.2. In Section 3B: ML for extracting functional forms, the authors mention that SR offers interpretability of the underlying physics. Could the authors help clarify this or mention specifically how Eqns. 4.1 and 4.2 can be interpreted? Thank you!

Response and Action: Thank you very much for your suggestion. For completeness and better understanding, we have added the following sentence in section 4(a) of the revised manuscript.

Thus, Equations (4.1) and (4.2) unfold the periodic behavior in combination with exponential growth or decay, describing the nature of the underlying force/moment variations in space and time.

4. (Table 3) For the varying number of beam Lagrangian points in Table 3, am I correct in believe that the grid resolution is also changing? Can the authors clarify this, either in the Table's caption or in the text, as well as provide what grid resolutions those numbers of beam nodes correspond to. Moreover, how are the beam's stiffness parameters being scaled in this study? Also, could

the authors clarify what the reference case is? Is this the higher resolution case mentioned earlier of 100×100 ?

Response and Action: Thanks for pointing out this. In the revised manuscript, we have added a note in the caption of Table 4, as follows.

Note: A consistent fluid domain grid resolution of (32×20) is used for different discretizations of the beam.

Also, we have included the procedure for calculating the beam stiffness in the Appendix, as mentioned below.

Stiffness Calculation:

In accordance with the fiber model reported in [30], the bending effect is realized by torsional springs (of stiffness K_B) connecting every three consecutive Lagrangian points, $(i-1)^{th}$, i^{th} and $(i+1)^{th}$, while the axial deformation is realized by linear springs (of stiffness K_S) linking every two consecutive Lagrangian points, i^{th} and $(i+1)^{th}$.

The bending stiffness, K_B in Equation (2.4), is computed according to the following procedure. For calculating K_B , a set of distinct external forces $\bar{\mathbf{f}}_j$ for $j = 1, 2, \dots, m$ are applied on the beam, which deflects all the nodes throughout it as per the following equation.

$$\bar{\mathbf{f}}_j = [K_G] \tilde{\boldsymbol{\psi}}_j . \quad (1)$$

In equation (1), $K_G \in \mathbb{R}^{2N \times 2N}$ is the global stiffness matrix; $\tilde{\boldsymbol{\psi}}_j \in \mathbb{R}^{2N \times 1}$ represents the displacement vector for all nodes, which is sparse everywhere except at the node onto which the force $\bar{\mathbf{f}}_j \in \mathbb{R}^{2N \times 1}$ has been applied. The lateral displacement of the i^{th} node is denoted by $\tilde{w}_{i,j}$ and is contained within $\tilde{\boldsymbol{\psi}}_{i,j}$. The bending stiffness is computed by using the slope of the force vs. nodal displacement. These stiffness values for all the nodes, $i = 1, 2, \dots, N$, are calculated according to the finite difference approximation, as follows.

$$K_{B_{i,j}} = \frac{\bar{f}_{i,j+1} - \bar{f}_{i,j}}{\tilde{w}_{i,j+1} - \tilde{w}_{i,j}} . \quad (2)$$

The axial stiffness, K_S in Equation (2.5), is computed directly as

$$K_S = \frac{(N-1)EA}{L} . \quad (3)$$

The bending stiffness and axial stiffness of the beam increase as the number of discrete nodes increases. The axial stiffness increases at a linear rate, and the bending (flexural) stiffness increases at a nonlinear rate.

Figure B: Flexural stiffness profiles for different FE discretizations: $N = 65, 100, 200$.